# Assessing the Transmissibility of the New SARS-CoV-2 Variants: From Delta to Omicron

**DOI:** 10.3390/vaccines10040496

**Published:** 2022-03-24

**Authors:** Rui Dong, Taojun Hu, Yunjun Zhang, Yang Li, Xiao-Hua Zhou

**Affiliations:** 1Yau Mathematical Sciences Center, Tsinghua University, Beijing 100084, China; dong-rui@tsinghua.edu.cn; 2Yanqi Lake Beijing Institute of Mathematical Sciences and Applications, Beijing 101408, China; 3Department of Biostatistics, School of Public Health, Peking University, Beijing 100191, China; 2011110158@stu.pku.edu.cn (T.H.); yunjun.zhang@pku.edu.cn (Y.Z.); 4Chongqing School, University of Chinese Academy of Sciences, Chongqing 400020, China; liyang@cigit.ac.cn; 5Beijing International Center for Mathematical Research, Peking University, Beijing 100191, China

**Keywords:** Delta variants, Omicron variants, B.1.617.2, AY.4, BA.1, reproductive number, vaccination, nonpharmaceutical interventions

## Abstract

Omicron, the latest SARS-CoV-2 Variant of Concern (VOC), first appeared in Africa in November 2021. At present, the question of whether a new VOC will out-compete the currently predominant variant is important for governments seeking to determine if current surveillance strategies and responses are appropriate and reasonable. Based on both virus genomes and daily-confirmed cases, we compare the additive differences in growth rates and reproductive numbers ($R_{0}$) between VOCs and their predominant variants through a Bayesian framework and phylo-dynamics analysis. Faced with different variants, we evaluate the effects of current policies and vaccinations against VOCs and predominant variants. The model also predicts the date on which a VOC may become dominant based on simulation and real data in the early stage. The results suggest that the overall additive difference in growth rates of B.1.617.2 and predominant variants was 0.44 (95% confidence interval, 95% CI: −0.38, 1.25) in February 2021, and that the VOC had a relatively high $R_{0}$. The additive difference in the growth rate of BA.1 in the United Kingdom was 6.82 times the difference between Delta and Alpha, and the model successfully predicted the dominating process of Alpha, Delta and Omicron. Current vaccination strategies remain similarly effective against Delta compared to the previous variants. Our model proposes a reliable Bayesian framework to predict the spread trends of VOCs based on early-stage data, and evaluates the effects of public health policies, which may help us better prepare for the upcoming Omicron variant, which is now spreading at an unprecedented speed.

## 1. Introduction

As new variants of SARS-CoV-2 (severe acute respiratory syndrome coronavirus 2) emerge, the accurate prediction of the next widespread variant is essential for early detection, vaccine research and policy design. Main variants in the past include the Alpha lineage (mainly B.1.1.7), Delta lineage (B.1.617.2 and all AY sublineages), and the most recent Omicron lineage (B.1.1.519 and all BA sublineages). The spread of a variant of concern (VOC) mainly depends on the comparative advantage in transmission relative to the predominant variant under natural selection. As the world is gradually recovering from the pandemic, it is crucial to estimate the transmissibility of VOCs when they appear, using data collected from the early stages, before making arbitrary decisions such as limiting transportation from a region to the rest of the world.

Alpha variant B.1.1.7 was first found in the United Kingdom (UK) in the Fall of 2020, and soon became predominant in most countries, especially the UK and United States (US) [1]. Local comparison between B.1.1.7 and the predominant variant B.1.177 suggested higher transmissibility of the former, and therefore, a selective advantage in the United Kingdom [2]. Researchers investigated time trends in the frequency of sampling Alpha variant genomes and concluded that B.1.1.7 was associated with a 50 to 100% higher reproductive number than the predominant variant, B.1.177, in the UK [2].

The dominant status of the Alpha variant lasted several months, until the detection of Delta variant, B.1.617.2, in the Winter of 2020. B.1.617.2 and its sublineages swept rapidly through India and the UK before reaching the US, where they quickly surged in numbers. In some US states, Delta has accounted for more than 99% of COVID-19 (coronavirus disease 2019) cases, and was leading to an overwhelming increase in hospitalizations [3]. As shown in Figure 1, since April 2021, the number of Delta cases recorded in the Global Initiative on Sharing All Influenza Data (GISAID) has increased dramatically, and therefore, a significant decrease of variants from other WHO (World Health Organization) labels, especially the predominant Alpha variants, was observed.

Most of the literature about COVID-19 vaccine development includes evolutionary analyses [4], antibody neutralizations [5], and protein structures [6]. However, few studies have examined the statistical trends from the early stages of spread, which would provide direct evidence of the transmissibility of a VOC and the effect of policies, such as the results concerning Alpha variants by Volz and colleagues [2]. So far, most research has been focused on the B.1.617.2 lineage as a representative of the Delta variants, but we noticed a global trend of a decreasing proportion of B.1.617.2 after May, especially in the UK, as shown in Figure 2, mainly due to the spread of AY.4, another Delta variant.

In November 2021, a new variant, Omicron, was identified in Africa. Since then, it has spread rapidly throughout Africa and Europe. Owing to this, it was possible to improve the model by Volz and colleagues [2] which illustrates the dominant role of Alpha through epidemiology data from 14 regions in the UK. In this research, we aimed to establish a modified model by combining both genomic and epidemiology data to address the issue of whether Omicron will out-compete Delta by evaluation its transmission ability.

In this paper, we propose an improved Bayesian model on the growth rate of a variant at a national level. We estimate the additive difference in growth rates and the ratios of the reproductive numbers of VOCs to their predominant variants, based on data collected during the early spread of those variants. Similar models can be found in [2,7,8]; however, our model considers two other sets of variables compared to [2]: interactions between countries, and common human interventions, including vaccinations and public transportation policies. As such, our model reflects reality in a more reasonable way. We applied the model to two Delta variants, B.1.617.2 and AY.4, in five major countries with the highest estimated case numbers in the world, as well as to recent Omicron variant data in the UK only, rather than in African countries where the variant first appeared. This is because data from Africa may suffer from a lack of reagent test kits and compliance and public trust regarding vaccines. In contrast, high-quality data from the UK can be obtained from public sources. Based on the early-stage data, i.e., when 1% to 10% of daily reported cases comprised the VOC, our model accurately predicted the date that this variant would become dominant (representing 50% of new infections) for the Alpha, Delta and Omicron variants.

## 2. Materials and Methods

### 2.1. Experimental Design

This study aimed to provide a universal overview of the increasing growth rate and reproductive number of novel prevalent variants of concern (VOC). Based on the method proposed by Volz and colleagues [2], we also considered international movement and various containment policies in certain countries, as both affect the growth rate of VOC. To our knowledge, this study is the first to provide a comprehensive analysis of the differences in the growth rates and reproductive numbers of the Delta and Omicron variants.

We downloaded the genomes of SARS-CoV-2 variants collected since 21 February 2021 from the GISAID public database. The two Delta VOCs, B.1.617.2 (period: 21 February 2021 to 17 July 2021) and AY.4 (period: 30 May 2021 to 24 July 2021), were selected based on their dominant roles in 2021. The period was determined according to the expansion process; it ended in August 2021, when Delta variants became dominant within the five selected countries and also worldwide, until the emergence of Omicron. BA.1 was chosen as the representative VOC of Omicron in the UK (period: 28 November 2021 to 10 December 2021). Based on the statistics of variants during a specific time period, a predominant variant is defined as the cocirculating non-VOC variant with the highest proportion in a given country.

### 2.2. Estimation of Number of Variants

Because of the different data disclosure policies in each country, data from GISAID cannot represent the actual case numbers. We estimated the numbers of variants based on both complete genome records on GISAID and daily reports from the Johns Hopkins University Center for Systems Science and Engineering (JHU CSSE) [9]. Under the assumptions that (1) submissions to GISAID are independent of variant information, and (2) JHU daily reports are accurate regarding total cases, the proportion of variants in each country is identical to those uploaded to GISAID, and thus, the number of variants can be estimated. Considering the fact that all users of GISAID agree that no restrictions should be attached to the submitted data, it is fairly safe to accept the first assumption. Regarding to the second, we selected JHU daily reports because they include cases confirmed by both national and state/local labs, in contrast to the data source from WHO and European Center for Disease Control and Prevention (ECDC). Nonetheless, the trends in JHU CSSE, WHO, and ECDC data are very similar [10].

We selected the five countries with the most estimated Delta variants, as shown in Table 1, and assessed the relationship between different countries based on international travel flows from Statista (https://www.statista.com, accessed on 30 July 2021). Communication among countries is important to the pandemic in many respects, for example, passengers can carry new variants to other countries by international travel. Therefore, the policies discussed in the following subsection should also be considered in the model. However, differences among the five countries, including population scales, predominant variants and vaccination levels may affect the results. To address this issue, we conduct an analysis of AY.4 based on data from the UK; the results coincided with the fact that AY.4 has outcompeted B.1.617.2 in the UK, as shown in Figure 2b.

### 2.3. NPIs and Vaccination

Data on NPIs and vaccinations were downloaded from the Oxford Coronavirus Government Response Tracker (OxCGRT) website by Ritchie and colleagues [10,11]. This database includes information on 13 nonpharmaceutical interventions (NPIs) and the vaccination rates for all countries. It tracks systematic information on policy measures that governments have applied to tackle COVID-19, covering over 180 countries. Policies are recorded on a scale to reflect the extent of government actions, and scores are aggregated into a suite of policy indices. The effects of policies are evaluated based on their parameters in the Bayesian model, focusing on differences regarding VOC compared to predominant variants. Specifically, the weekly additive difference in growth rates is sampled from the normal distribution with mean as the difference in the previous week, plus the effects of 13 NPIs and vaccinations in the previous weeks (see Equation (4) in Section 2.4).

### 2.4. Bayesian Model and Phylo-Dynamics Analysis

Among the five countries, we sequentially completed the following analysis to estimate the relative advantage of the VOC against the predominant variant for each country. The additive difference of growth rate between the VOC and predominant was estimated based on an improved Bayesian framework by considering both the correlation among countries and the effect of NPIs and vaccinations, compared to the model by Volz and colleagues [2]. We defined the count of VOC genomes as $B_{i,w}$, out of a total count of selected viral sequences $T_{i,w}$ for week w and country i. $B_{I,w}$ follows a binomial distribution with a time-varying probability for each country, where the probability is modelled through a logistic linear model:(1)$$\begin{array}{c} B_{i,w}∼\mathrm{Binom}\left(T_{i,w},p_{i,w}\right)\\ \mathrm{logit}\left(p_{i,w}\right)=\phi_{i,w} \end{array}$$

We assume that the log odds of sampling frequency for VOC $\phi_{i,w}$ may be controlled by a Markov Chain, where
(2)$$\phi_{i,w+1}=\phi_{i,w}+\rho_{i,w}$$

Variable $\rho_{i,w}$ represents the weekly growth rate in frequency of VOC, suggesting the additive difference in growth rate for the VOC compared with other lineages. We model $\rho_{i,w}$ from a multivariate normal distribution with nonidentity variance–covariance matrix Σ. The variance and covariance are estimated based on data of international travel inflows and outflows, collected from Statista (https://www.statista.com, accessed on 30 July 2021). The principal diagonal entries are scaled to 0.1, similar as in the method proposed by Volz and colleagues [2]. Assuming that the total number of countries are denoted as K,
(3)$$\begin{array}{c} \rho_{w}=\left(\rho_{1,w},\rho_{2,w},\cdots ,\rho_{K,w}\right)\\ \rho_{w}∼N\left(\mu_{w},\Sigma \right), \end{array}$$

The 13 NPIs and vaccinations are represented as covariates ${\mathrm{cov}}_{i,w}$ functioning on the average additive difference in growth rate, $\mu_{i,w}$
(4)$$\mu_{i,w+1}∼N\left(\mu_{i,w}+\beta^{t}{\mathrm{cov}}_{i,w},\sigma_{t}\right)$$

Priors are modelled in a similar manner:(5)$$\begin{array}{c} \phi_{i,1}∼N\left(-7,10\right)\\ \sigma_{t}∼\mathrm{Exp}\left(100\right)\\ \beta ∼N\left(0_{p},I_{p}\right) \end{array}$$

A Bayesian coalescent phylo-dynamic analysis was then performed with BEAST v1.10.4 (Bayesian Evolutionary Analysis Sampling Trees) [12]. From this result, we also obtained estimations of clock rates. The clock rate here serves to estimate the number of substitutions per site per year. A phylogenetic tree was constructed using the maximum likelihood method in iQtree v2.1.3 [13] assuming a HKY model (Hasegawa-Kishono-Yano model) for substitution. The selection of the HKY model was based on the fact that the substitution rates of transitions are higher than expected by chance relative to those of transversions in SARS-CoV-2, which makes the Jukes-Cantor model incompatible in this case [14,15]. Based on the estimated additive differences in growth rates and clock rates for each variant, the ratio of reproductive number was calculated using the “mlesky” and “treedater” packages in the R software, v4.1.2. The reproductive number was calculated according to its definition and estimated according to the time-varying effective size. The estimated ratio of reproductive number suggests the transmission advantage of a VOC against compared to variants. We denote the growth rate of hosts carrying non-VOC as r(t), and those carrying VOC as $r^{\prime }\left(t\right)$. Both r(t) and $r^{\prime }\left(t\right)$ can be obtained from the estimated effective size, and the corresponding time-varying reproductive number of non-VOC and VOC are denoted as R(t) and $R^{\prime }\left(t\right),$ respectively. According to the definition of reproductive number, we can estimate the ratio of the reproductive number of a given VOC against the non-VOC as
(6)$$\phi_{t}^{1}=\frac{R_{t}^{\prime }}{R_{t}}=\frac{\int_{0}^{+\infty }g\left(s\right)e^{-\int_{0}^{s}r\left(t\right)dt}}{\int_{0}^{+\infty }g\left(s\right)e^{-\int_{0}^{s}\left[r\left(t\right)+\rho \left(t\right)\right]dt}}$$
where g(s) is the density of generation time. Utilizing the same token, we can also estimate the ratio through $r^{\prime }\left(t\right)$:(7)$$\phi_{t}^{2}=\frac{R_{t}^{\prime }}{R_{t}}=\frac{\int_{0}^{+\infty }g\left(s\right)e^{-\int_{0}^{s}\left[r^{\prime }\left(t\right)-\rho \left(t\right)\right]dt}}{\int_{0}^{+\infty }g\left(s\right)e^{-\int_{0}^{s}r^{\prime }\left(t\right)dt}}$$

A robust estimator of the ratio can be obtained by averaging the above two estimations, denoted as
(8)$$\hat{\phi_{t}}=\frac{\phi_{t}^{1}+\phi_{t}^{2}}{2}$$

Additionally, we conducted a simulation based on the same statistical model and estimated the date when a VOC would become dominant (proportion ≥ 50%). To this end, we first estimated the posterior distribution of parameters given in the logistic linear model under a Bayesian framework and then simulated the growth curve of the proportion of VOC for 10,000 independent trails. Details of the model can be found in the Appendix A.

## 3. Results

### 3.1. Global Trends and Major Countries

As Figure 1 and Figure 2 show, we selected the two main Delta variants, B.1.617.2 and AY.4, at their respective early stages of transmission (21 February 2021 to 17 July 2021, and 30 May 2021 to 24 July 2021) as VOCs with the corresponding predominant variant in each country during those periods. When we considered B.1.617.2, we selected the predominant variant in each country for comparison purposes. When AY.4 became a VOC, we found that the predominant variant was B.1.617. In all analyses, we compared the VOC with its predominant variant. The number of VOCs was estimated for all countries. In Table 1, we present data for the five countries with highest numbers of estimated VOCs: India, Indonesia, Russia, the United Kingdom, and the United States. The index in Table 1 indicates the proportion of complete genome records on GISAID to the daily reports on JHU CSSE; the UK contributes to the largest extent in this regard.

We also collected the Omicron lineages in the UK from 28 November 2021 to 10 December 2021 to estimate the difference in growth rate of that VOC and the corresponding predominant variant.

### 3.2. Relative Transmission Advantage of B.1.617.2 Compared to Predominant Variants

As shown in Figure 3a, the overall global additive weekly difference of B.1.617.2 compared to predominant variant in terms of growth rate was initially 0.44 (95% CI: −0.38, 1.25) in late February. It then reached a peak in succession followed by a drop-off, i.e., −0.17 (95% CI: −0.69, 0.36) in mid-July. The average advantage over the whole period was the largest in the US and UK, as shown in Figure 3b, with medians of 0.43 (95% CI: −0.02, 0.88) and 0.39 (95% CI: −0.10, 0.87), respectively. In contrast, the differences in Indonesia ranged from −0.38 (95% CI: −0.82, 0.04) to 0.89 (95% CI: 0.46, 1.36), with the smallest advantage being observed with B.1.617.2.

The results in Figure 4a suggest that B.1.617.2 had the highest clock rate in Indonesia compared to other countries, representing the smallest difference in $R_{0}$ between B.1.617.2 and the predominant variant in Figure 4b. The clock rate here served to estimate the number of substitutions per site per year, as defined and calculated in BEAST based on the HKY model. The reason for the high clock rate of B.1.617.2 in Indonesia might be related to its late appearance and the presence of various predominant variants. Figure 5 displays the ratio of $R_{0}$ between B.1.617.2 and its predominant variant, where the global ratio (dotted line in Figure 5b) shows an inverted U curve, while the average ratio over the period was greater than 1 for all five countries (Figure 5a). In the US, the weekly ratio reached a peak of 2.22 (95% CI: 1.46, 3.44) in early June, with the highest $R_{0}$ of B.1.617.2 being 1.48 (95% CI: 0.74, 2.65). The lowest mean ratio of $R_{0}$ was observed for Indonesia, i.e., 1.09 (95% CI: 0.56, 2.25).

In Figure 6, we estimated the origin date of B.1.617.2 in each country. It is likely that this variant originated in India, the country in which the Delta variant was first identified, albeit much earlier.

### 3.3. Relative Transmission Advantage of AY.4 compared to B.1.617.2

We conducted an analysis of AY.4 in the same manner, i.e., based on the records collected from the UK on GISAID. Figure 7a displays the clock rates of AY.4 and B.1.617.2 in the UK, suggesting that AY.4 mutated more rapidly. As Figure 7b shows, the ratio of $R_{0}$ was slightly over 1 (mean: 1.01 (95% CI: 0.56, 1.89)), suggesting a slight advantage of AY.4 over B.1.617.2 in the UK.

### 3.4. Differences in Growth Rates between Alpha, Delta and Omicron

Figure 8 shows the differences in growths rate between VOC and their predominant variants in the UK during the main outbreak in 2021. Over 99% of Omicron variants collected between 28 November 2021 and 10 December 2021 were BA.1, and the additive difference in growth rates reached 2.66 (95% CI: 1.44, 3.87) for BA.1 compared to AY.4. The difference of B.1.617.2 to B.1.1.7 was about 0.39 (95% CI: −0.1, 0.87), followed by a slight positive advantage of AY.4 over B.1.617.2 (0.04, 95% CI: −0.53, 0.61).

### 3.5. Effect of NPIs and Vaccination

For the 13 nonpharmaceutical interventions (NPIs) used in previous research [10,11], we defined controlling efficiency as the additional control power regarding a VOC compared to its predominant variant. Negative values suggest a lower ability to control a VOC. We found two aspects with negative values, i.e., COVID-19 testing policy (mean: -0.64 (95% CI: −1.09, −0.19)) and controlling international travel (mean: −1.68 (95% CI: −2.31, −1.03)), two with positive values, namely, public campaigns and advocating the use of masks. No significant differences were observed for other NPIs or vaccination rates as shown in Table 2, suggesting that vaccinations remained effective against Delta variants compared to predominant variants.

### 3.6. Estimating the Date on which a VOC Becomes Dominant

While attempting to estimate if and when a VOC will become dominating, we found that most variants, such as Beta and Mu, failed to reach to 1% of the daily collection on GISAID. On the other hand, Alpha, Delta and Omicron all reached almost 100% in the end, albeit at different rates. Therefore, a possible strategy for governments could be to focus on VOCs that make up over 1%. As the results in Figure 9 suggest, our predictions regarding Alpha, Delta and Omicron were similar to the real data. For Alpha, the estimated date on which that VOC would become dominant the UK was 27 December 2020 (90% CI: 13 December 2020, 31 January 2021). In reality, on the week of 20 December 2020, B.1.1.7 comprised o over 50% of the daily reports. Considering that Omicron is spreading much faster than the other two VOCs, we estimated the precise date, rather than the week, on which it would become dominant: 14 December 2021 (90% CI: 11 December 2021, 29 December 2021). In fact, over 50% of daily collected samples since 12 December 2021 are Omicron.

## 4. Conclusions

Since the COVID-19 pandemic started in 2020, the accuracy of assessments of variants has played a crucial role in our understanding of the spread of the disease. It is of great importance to analyze the Omicron variant based on our experience with Delta. Therefore, we propose a Bayesian framework to estimate the transmissibility of SARS-CoV-2 variants of concern (VOC) compared to their predominant variants to determine whether Omicron will out-compete Delta. In contrast to the related literature by Volz and colleagues [2], we also considered communication among major countries and the pandemic policies in each country, as these factors reflect reality in a more reasonable way. Our model can be used to estimate differences in growth rates for VOCs compared with predominant variants, and therefore, the reproductive numbers to describe the transmissibility of VOCs and future spread trends. The model also evaluates the effect of public health policies, which may help us better prepare for future Omicron variants. We found that Delta variants spread more rapidly, even in the early stages. Our result suggests that Omicron is spreading at an unprecedented rate.

## 5. Discussion

Our choice of the five major countries which were the focus of this study (see Table 1) was based on their data from GISAID and JHU CSSE, under the assumption that the proportion of variants is independent from sequencing policies and submission. The index suggests that developed Western countries, such as the UK and US, tend to contribute more records to public databases, and predictions based on such data are more reliable and robust. India, Indonesia, and Russia submit no more than 0.2% of their daily report sequences to GISAID. It is important to point out that our project is data-driven; thus, the reliability of the data sources is of fundamental importance. In this study, we did not examine the deficiencies and biases that are intrinsic to such data. However, all three datasets are widely used in the literature on COVID-19 [16,17,18,19,20,21,22,23,24,25,26,27,28,29], and are collected by authorities such as national governments and departments of public health. Certain issues regarding these data are taken into consideration in our model; for example, lags in submissions are addressed by removing sequences which are too recent or which show abnormal decreases. We also conducted a sensitivity analysis based on JHU CSSE daily case numbers. The result showed that our model was robust regarding to total case numbers (details in the Appendix A). If any better data source becomes accessible in the future, the results based on our model could be further improved.

First, in the comparison between B.1.617.2 and predominant variants in Figure 3, Figure 4 and Figure 5, the advantage of B.1.617.2 is most obvious when compared to Alpha variants (B.1.1.7 in the UK and US) with regard to the predominating variants in other countries. It is important to point out that this advantage is relative, rather than a characteristic of the B.1.617.2 variant alone. This may explain the fact that Delta put an end to the Alpha era and dominated the world in a few months. We also estimated the origin date of B.1.617.2 in each country, as shown Figure 6; consistent with the first record of B.1.617.2 on GISAID, the very first B.1.617.2 sequence was estimated to have come into existence in February 2020, i.e., almost one year before its large-scale spread. Combined with the index in Table 1, our model is reliable when the data quality is high. In the UK and US, the estimation and submitted sates on GISAID were only few days apart, but in Russia, these dates differed by several months.

The drop in the proportion of B.1.617.2 since June 2021 is mainly due to the presence of AY.4, especially in the UK. When AY.4 appeared, the world was dominated by B.1.617.2. The advantage which AY.4 has over B.1.617.2 lies in its growth rate and reproductive number; both are Delta variants which share important mutations such as D614G, T478K, P681R and L452R [30,31,32,33,34]. An atypical deletion of six nucleotides on AY.4 may cause N gene target failure (NGTF) and its slight advantage, based on a case of a 59-year-old, unvaccinated Italian male [33]. Figure 7b reveals the gradual dominant status of AY.4 within the Delta family in the UK. A comparison between AY.4 and B.1.617.2 in other countries such as Pakistan and Denmark also suggests similar outcomes [32,35].

Regarding the recent Omicron variant, due to the fact that its prevalence is still limited to certain countries, the identification of countries with the highest numbers of Omicron variants requires a large quantity of data. Though it was first found in Africa, local testing data are inaccurate due to a lack of reagent test kits and poor compliance among the population. The populations of the UK and South Africa in 2020 were roughly 67 and 59 million respectively; however, the daily testing numbers on 1 December 2021 were 1.8 million versus 35,000, suggesting a significant shortage of testing infrastructure in South Africa. Therefore, we only selected variants in the UK (collected between 28 November 2021 to 10 December 2021) to study the growth rate of Omicron. Four Omicron variants are found in the UK: B.1.1.529 (the main variant in South Africa), BA.1, BA.2 and BA.3. However, out of all 5952 Omicron variants in the UK, 5947 were identified as BA.1 which was therefore chosen as the representative of Omicron variant. The predominant variant in the UK for this period was AY.4, and comparisons between AY.4 and BA.1 can be considered as representative of the differences between Omicron and Delta. The results in Figure 8 suggest the tremendous advantages that Delta has based on current evidence.

Regarding our policy evaluations, based on the controlling efficiency of 13 NPIs and vaccination to Delta variants, current policies, i.e., public campaigns and advocating for the use of facial coverings are still effective, especially in terms of lowering the ratio of incidence for B.1.617.2 transmissions against transmissions triggered by the predominant variant. However, current testing policies and restrictions on international travel were found to affect the spread of Delta variants in a less efficient way compared to the Alpha era [36,37,38]. Similar research has suggested that existing public health measures, including mass testing strategies, were effective but insufficient to control the outbreak of the Delta variant in Vietnam [38]. Vaccination remains as effective for Delta variants as it is for Alpha variants, but this is unclear for Omicron. Current evidence for Omicron consistently demonstrates that the variant will become dominant within weeks. As such, we recommend a stricter response to prevent Omicron spread. Please note that this suggestion is based on our history about Delta, rather than direct evidence of Omicron. There is not yet reliable empirical data to estimate the effects of policy on the transmission Omicron variants, as the effect can only be observed several weeks after the fact. Thus, the current data still reflect different transmissibility of Omicron and the predominant Delta variants.

Finally, our model also estimates when a VOC will become dominant, using data of its early-stage spread. This was applied to the Alpha, Delta and Omicron variants. Accurate estimations enhance our understanding of VOCs and allow us to better prepare in the early stages. Based on past experience, Beta, Gamma and other variants failed to reach 10% of the daily samples on GISAID in the country which fist collected them, and as such, did not become dominant. Regarding the main global threats so far, i.e., Alpha, Delta and Omicron, our prediction agrees with real-world statistics during the spread of each VOC in the UK.

While our proposed model effectively estimates the trends of VOCs compared to their predominant variants, it can be improved upon in the following aspects. First, the predominant variant can be extended to multiple predominant variants. For example, the cocirculating Delta variants in the UK of Omicron consisted of AY.4 (56.21%), AY.4.2 (15.75%) and AY.4.2.2 (6.51%), among others. What Omicron faces is a mixture of all cocirculating variants rather than just one, i.e., AY.4. Thus, comprehensive comparisons could be more accurate. Second, the accuracy of the model relies on the accuracy and transparency of the statistical data. When the data do not correspond to the actual situation, conclusions will be insufficient, as was the case in our estimations for Russia. The reason of this insufficiency in Russia is mainly the lack of sequences in the public database (GISAID), as shown in Table 1. Of the estimated > 1 million records between February and July in 2021, only about 1000 complete sequences were uploaded to GISAID. Therefore, conclusions for countries with low indices may not reflect the reality. Third, our understanding of the pandemic is based largely on our experiences with Delta and responses to it. Thus, complex analyses might attempt to model the factors specific to Omicron, such as the local situation in Africa and the flight ban between Africa and other countries announced in November/December 2021. The situation in this regard might be closer to the lockdown in Wuhan during the outbreak in early 2020. Finally, our model does not consider the intrinsic fluctuation dynamics and the parameters determining those dynamics. Our model avoids fluctuations in the initial stages by avoiding the days after the first appearance of the VOC in question; however, the prediction model could be improved substantially if it could use data regarding a VOC from the time of its first appearance. Additionally, the factors affecting the spread of SARS-CoV-2 [39,40,41], such as vaccination rates, should be taken into consideration; in some countries, a large proportion of people have been fully vaccinated as of early-2022, leading to an entirely different environment compared to that of early 2021. Other epidemiology research may apply the SIR (susceptible-infected-recovered) and SEIR (susceptible-exposed-infected-recovered) models to address this problem. One possible approach might be to combine our Bayesian model with the SIR model to address this issue [42,43,44]. Nonetheless, we hope that our estimates will provide a useful framework for assessments of the transmissibility of VOCs when they first appear, and will provide guidance for governments and individuals seeking to respond in reasonable ways.

## Figures and Tables

**Figure 1 vaccines-10-00496-f001:**
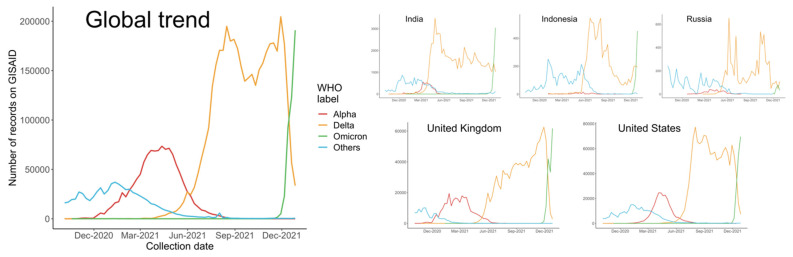
Number of sequences on GISAID collected from October 2020 to December 2021 globally and in each country.

**Figure 2 vaccines-10-00496-f002:**
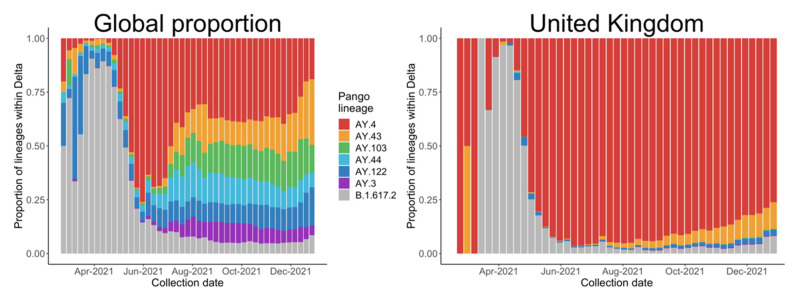
Proportion of the main Delta variants collected from 22 February 2021 to 31 December 2021, globally and in the United Kingdom.

**Figure 3 vaccines-10-00496-f003:**
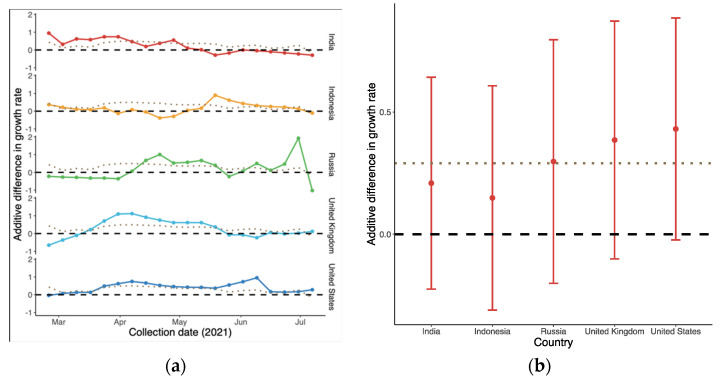
Additive difference in growth rate of B.1.617.2 compared to the predominant variants in each country: (**a**) Time-varying trend; and (**b**) Average difference in growth rate during the period. The dotted line represents the global trend for comparison. The dashed line in (**b**) indicates that there was no difference between the two variants.

**Figure 4 vaccines-10-00496-f004:**
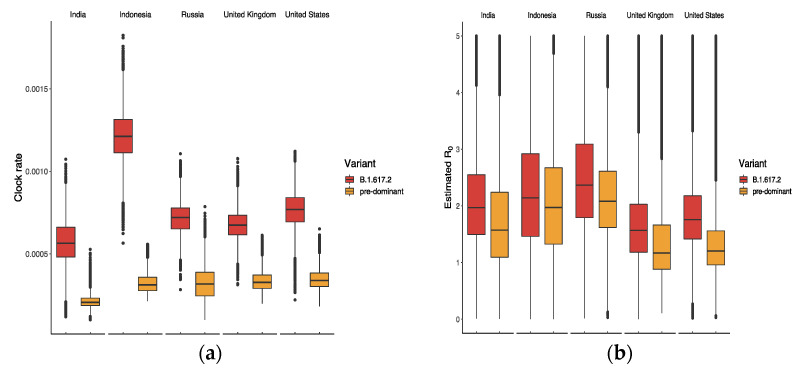
Estimated values of (**a**) clock rate and (**b**) reproductive number of B.1.617.2 and its predominant variants in five countries.

**Figure 5 vaccines-10-00496-f005:**
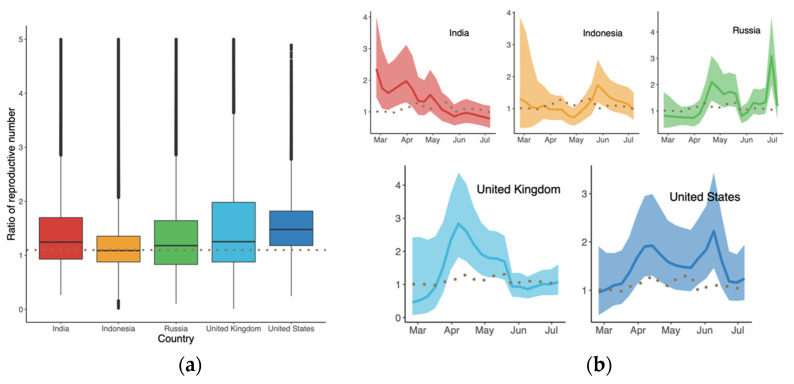
Ratio of the reproductive numbers of B.1.617.2 and predominant variants (**a**) over the indicated period, and (**b**) time-varying trend for each country. Dotted lines represent the global level.

**Figure 6 vaccines-10-00496-f006:**
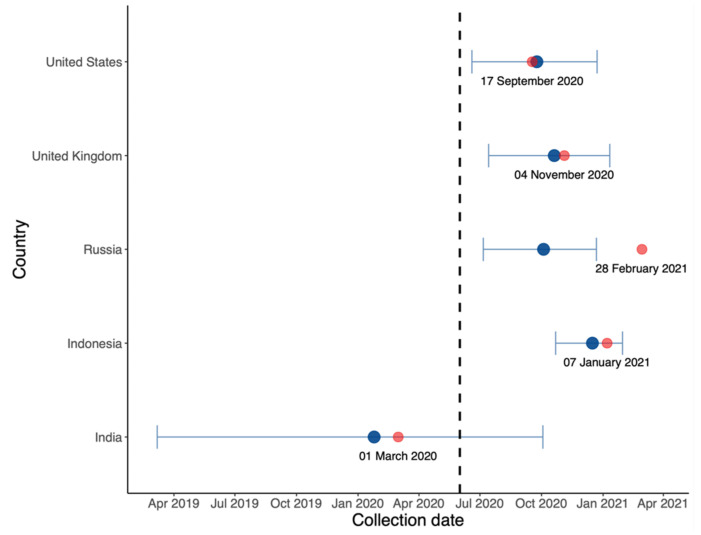
Estimated origin date (blue) and collection date (red) of the first B.1.617.2 sequence in each country on GISAID with annotated collection date.

**Figure 7 vaccines-10-00496-f007:**
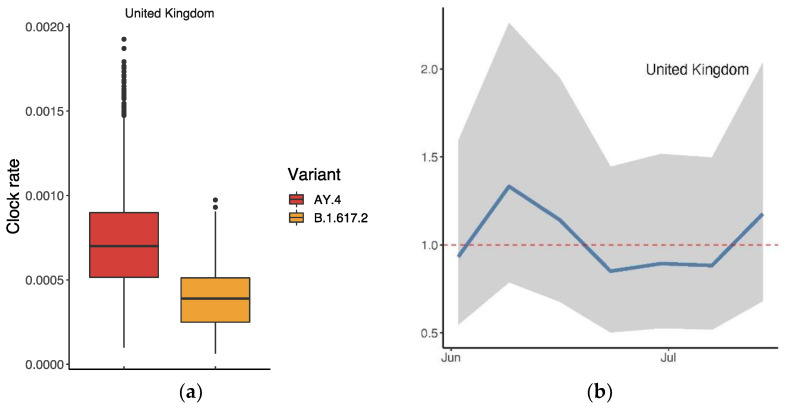
Comparison between AY.4 and B.1.617.2 in the UK: (**a**) Ratio of the average reproductive numbers of AY.4 and B.1.617.2 over the indicated period; (**b**) time-varying trend of the ratio. The dashed line indicates when there was no advantage (ratio = 1).

**Figure 8 vaccines-10-00496-f008:**
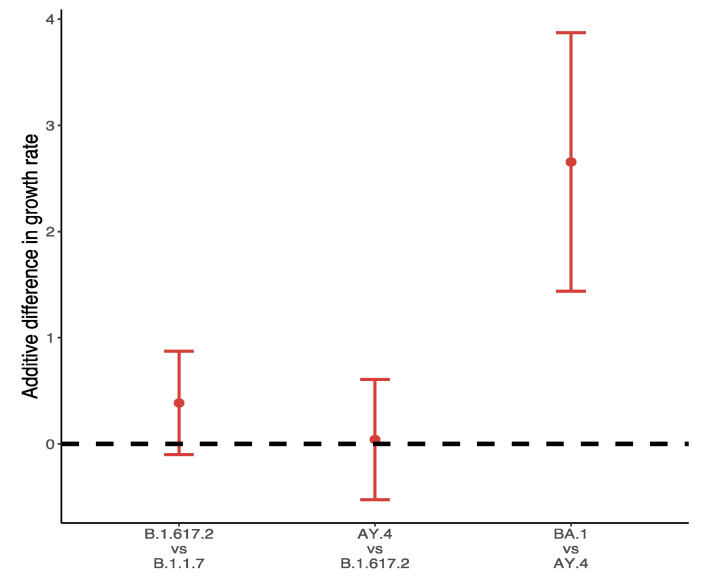
Difference of weekly growth rates between VOC and its predominant variant.

**Figure 9 vaccines-10-00496-f009:**
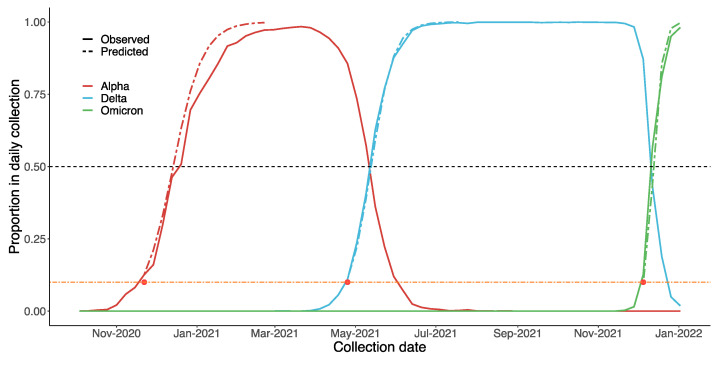
Estimated spread trends of Alpha, Delta, and Omicron in the UK using the Bayesian model, based on GISAID data. The black dashed line suggests when each variant will become dominant (proportion > 50%). Predictions were based on data from the early stages of each VOC (proportion < 10%, below the orange dashed line). On this basis, we predicted the trends of these VOCs after they had reached 10% (after the red dot, i.e., above the orange dashed line).

**Table 1 vaccines-10-00496-t001:** The five countries with the largest numbers of estimated B.1.617.2 and AY.4 variants.

	n_GISAID ^1^	n_estimated ^2^	Name	n_GISAID ^1^	n_estimated ^2^	n_estimated ^2^	Index
**1 Februay 2021 to 17 July 2021**	**B.1.617.2**		**predominant variant**			**all**	
**India**	$1.57\times 10^{4}$	$1.37\times 10^{7}$	B.1.617.1	$3.89\times 10^{3}$	$2.56\times 10^{7}$	$2.01\times 10^{7}$	0.0015
**Indonesia**	$9.79\times 10^{2}$	$5.55\times 10^{5}$	B.1.466.2	$1.19\times 10^{3}$	$5.63\times 10^{5}$	$1.56\times 10^{6}$	0.0021
**Russia**	$1.23\times 10^{3}$	$8.48\times 10^{5}$	B.1.1.523	$3.69\times 10^{2}$	$1.93\times 10^{5}$	$1.75\times 10^{6}$	0.0019
**United Kingdom**	$1.20\times 10^{5}$	$5.17\times 10^{5}$	B.1.1.7	$1.42\times 10^{5}$	$5.77\times 10^{5}$	$1.29\times 10^{6}$	0.2462
**United States**	$3.39\times 10^{4}$	$6.09\times 10^{5}$	B.1.1.7	$2.04\times 10^{5}$	$2.82\times 10^{6}$	$5.95\times 10^{6}$	0.0725
**30 May 2021 to 24 July 2021**	**AY.4**		**predominant variant**			**all**	
**India**	$1.57\times 10^{4}$	$4.00\times 10^{5}$	B.1.617.2	$3.52\times 10^{3}$	$2.47\times 10^{6}$	$3.48\times 10^{6}$	0.0014
**Indonesia**	$4.56\times 10^{2}$	$4.12\times 10^{5}$	B.1.617.2	$9.00\times 10^{2}$	$7.04\times 10^{5}$	$1.32\times 10^{6}$	0.0013
**Russia**	$1.70\times 10^{1}$	$1.48\times 10^{4}$	B.1.617.2	$8.95\times 10^{2}$	$5.99\times 10^{5}$	$8.65\times 10^{5}$	0.0015
**United Kingdom**	$5.59\times 10^{4}$	$4.85\times 10^{5}$	B.1.617.2	$1.10\times 10^{5}$	$7.22\times 10^{5}$	$1.20\times 10^{7}$	0.1527
**United States**	$1.36\times 10^{4}$	$2.04\times 10^{5}$	B.1.617.2	$3.99\times 10^{4}$	$5.10\times 10^{5}$	$1.18\times 10^{6}$	0.0782

^1^ Number of sequences submitted to GISAID. ^2^ Estimated number of variants based on GISAID and JHU CSSE.

**Table 2 vaccines-10-00496-t002:** Estimate of the effects of 13 NPIs and vaccination.

Intervention	Effective Size	95% CI
Covid 19 testing policy	−0.64	(−1.09, −0.19)
Covid contact tracing	0.04	(−0.3, 0.39)
Covid vaccination policy	0.01	(−0.17, 0.19)
Debt relief	−0.07	(−0.28, 0.14)
Face covering policies	0.42	(0.05, 0.78)
Income support	−0.09	(−0.48, 0.29)
International travel	−1.68	(−2.32, −1.03)
Public campaigns	2.24	(1.07, 3.4)
Public events	0.26	(−0.1, 0.63)
Public gathering rules	0.32	(−0.13, 0.76)
Public transport	−0.2	(−0.5, 0.11)
School closures	0.05	(−0.25, 0.35)
Stay at home	−0.13	(−0.46, 0.2)
Workplace closures	−0.32	(−0.69, 0.05)

## Data Availability

The research is mainly based on the public data on GISAID (https://www.gisaid.org, accessed on 12 February 2022) and JHU CSSE (https://github.com/CSSEGISandData/COVID-19, accessed on 12 February 2022). Data on NPIs and vaccination are downloaded from the website (https://ourworldindata.org/policy-responses-covid, accessed on 1 December 2021) [10,11]. We assess the communication of different countries with the international traveling flows from Statista (https://www.statista.com, accessed on 30 July 2021).

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
