# Peer review of "Assessing the Transmissibility of the New SARS-CoV-2 Variants: From Delta to Omicron"

_vaccines, 2022, doi:10.3390/vaccines10040496_

Round 1

Reviewer 1 Report

Dong, et al. compared the transmissibility of the SARS-CoV-2 variants including Delta strains B1.617.2 and AY.4 in India, Indonesia, Russia, United Kingdom, and United States using the methods previously proposed by Volz, et al. Authors estimated the spread trend of different variants including Alpha, Delta, and Omicron on Bayesian model based on published SARS-CoV-2 sequence database, and provided potential models for the evaluation of transmissibility of the SARS-CoV-2 variants.

Comments:

  1. The virus the new variants of “SARS-CoV-2” needs to be indicated in the article title.
  2. Figure 1 and 2 displayed the sequence numbers and proportion of SARS-CoV-2 variants in global but provided only limited information for the studied countries (India, Indonesia, Russia, United Kingdom, and United States). Display the sequence numbers and proportion of SARS-CoV-2 variants in the studied countries in Figure 1 and 2 will more clearly show the trend of the variants in these countries.
  3. Page 3, line 102-103. Why the collection periods for the sequences of two Delta variants stopped in July 2021? Did the Delta variants become predominant strains in the five studied countries in July 2021?
  4. Page 4, line 122-124. Please describe method that evaluated the effect of NPIs and vaccination.
  5. Page 4, line 134. Please describe how to select HKY model for substitution.
  6. Page 5, line 157-159. Figure 3a did not show the global data of additive weekly difference. Please add the global data in the Figure 3a and 3b, as well as Figure 4a and 4b in Page 6. Similar revisions were needed in Figure 8a to 8c. Also, Figure 8c is blank.
  7. Page 6, line 167. Please define the substitution rate and clock rate and describe how to estimate the values in studied countries.
  8. Page 6, line 168. The method that estimated R0 value in Figure 4b needs to describe in Methods section.
  9. Page 6.
  10. Page 6, line 170. In addition to the global ratio in Figure 5a, add the curves for the studied countries. Also, add the global value in Figure 5b and 5c.
  11. Page 7, Figure 5c is blank.
  12. Page 8, line 191. Why only three countries were selected as analyzing AY.4 variants?
  13. Page 10, line 212. Please indicate all the estimated values of the effects of all 13 NPIs and vaccination in an additional table.
  14. Page 11, Figure 10. Please display the Delta variants data in UK to make sure the consistent display of all variants in this Figure. The method to predict the spread trend was needed to describe in Method section.
  15. Page 11, line 268-270. The description needs reference citations.
  16. Page 11, line 270-271. The difference of clock rates in Figure7 and Figure 4a were not obvious.
  17. Page 12, line 291-292. The description needs reference citations.
  18. Page 13, line 326. Please indicate which result of Russia had limitation and describe the reason that cause limitation of the data from Russia.

Author Response

Firstly we would like to thank you for reviewing our manuscript, and we benefit a lot from the revising process based on your suggestions. Please see below for our responses and when the line number is mentioned, it refers to the revised manuscript version with marked notes (not the clean revised version).

Comments:

  1. The virus the new variants of “SARS-CoV-2” needs to be indicated in the article title.

    Response: Thanks for pointing this out. We have added the "SARS-CoV-2" in the title, and the new title is "Assessing the transmissibility of the new SARS-CoV-2 variants: from Delta to Omicron". We agree that it is better illustrated the topic in this way as the journal publishes papers in various topics.

  2. Figure 1 and 2 displayed the sequence numbers and proportion of SARS-CoV-2 variants in global but provided only limited information for the studied countries (India, Indonesia, Russia, United Kingdom, and United States). Display the sequence numbers and proportion of SARS-CoV-2 variants in the studied countries in Figure 1 and 2 will more clearly show the trend of the variants in these countries.

    Response: Thanks for your suggestions. We have added the trend in each country in Figure 1 next to the global trend, and due to the different scales, we split into countries instead of drawing everything in one figure. Regarding to Figure 2, we add the proportions of variants in the United Kingdom, where AY.4 has the most significant advantage over all. Besides, we also update the results in these figures to the end of 2021 so that the result is more up-to-date. The reason that we eliminate data from 2022 is that the testing policy in several countries has changed (e.g., no free testing to public in UK), thus the data would may have a completely different structure.

  3. Page 3, line 102-103. Why the collection periods for the sequences of two Delta variants stopped in July 2021? Did the Delta variants become predominant strains in the five studied countries in July 2021?

    Response: Yes, the Delta variant becomes the dominating variant in all those five countries, and actually in most countries in the world, until the emergence of Omicron in November 2021. Then Delta variants becomes the "pre-"dominant strains. For the B.1.617.2, it is the first Delta variant that becomes globally popular, and during July 2021, from Figure 1, it has already out-competed the pre-dominant variants including Alpha. And for AY.4, we mainly study its transmissibility advantage within the Delta family, i.e., B.1.617.2. This trend is illustrated in Figure 2, where B.1.617.2 takes a large proportion since February 2021, decreases till July 2021 and becomes stable afterwards globally and especially in the UK. In other words, the emergence and expansion of AY.4 will be during the period of June/July 2021.

    To clarify this, in the revised manuscript, we add "The period is determined according to their expansion process, and stopped at August 2021, when the Delta variants becomes dominating within the five selected countries and also worldwide until the emergence of Omicron"in Section 2.1 (revised manuscript: line 117-119).

  4. Page 4, line 122-124. Please describe method that evaluated the effect of NPIs and vaccination.

    Response: Thanks for pointing this out. To better illustrate our model, we re-write Section 2.4, and regarding to NPIs and vaccinations, we also emphasize in Section 2.3 (revised manuscript: line 144-146) that "Specifically, the weekly additive difference in growth rate is sampled from a normal distribution with mean as the difference in the previous week plus the effects of 13 NPIs and the vaccinations in the past weeks (see equation (4) in Section 2.4)" in Section 2.3. In the previous version the mathematical details were all in the Supplementary Information but more than one reviewers found the description in Section 2.4 (previous title: Statistical analysis) is too short and vague. Thus we move the major model back to the content, and hope that the current version is better for readers to understand.

  5. Page 4, line 134. Please describe how to select HKY model for substitution.

    Response: We select the HKY model instead of the JC model as the substitution rates of transitions and transversions are usually different especially in RNA viruses, and in SARS-CoV-2 viruses, a strong bias in transition versus transversion ratio is observed [14,15]. And we also make it clear in the manuscript by adding the following content "The selection of HKY model is based on the fact that the substitution rates of transitions are higher than expected by chance relative to those of transversions in SARS-CoV-2, which therefore makes the Jukes-Cantor model incompatible in this case [14,15]" in Section 2.4 (revised manuscript: line 180-183).

  6. Page 5, line 157-159. Figure 3a did not show the global data of additive weekly difference. Please add the global data in the Figure 3a and 3b, as well as Figure 4a and 4b in Page 6. Similar revisions were needed in Figure 8a to 8c. Also, Figure 8c is blank.

    Response: As you suggested, we add the results of the global data and add the results in Figure 3 and Figure 5 (revised manuscript: line 231-235, line 254-257). In Figure 7 (the previous Figure 8) (revised manuscript: line 280-285), to simplify, we keep only the results globally and in the UK. We think it is also necessary to point out that conducting analysis on the global level is not very informative because different countries have different policies against the pandemic on the same days. Thus the global level here is actually an average of the countries, rather than consider all countries together as a whole.

  7. Page 6, line 167. Please define the substitution rate and clock rate and describe how to estimate the values in studied countries.

    Response: Thanks for your suggestion. Firstly, in Figure 4a, we aim to compare the clock rate (rather than substitution rate) of B.1.617.2 to the pre-dominant variants, and show that B.1.617.2 mutates more often. We add the definition that "The clock rate here is to estimate the number of substitutions per site per year as defined and calculated in BEAST based on HKY model" in Section 3.2 (revised manuscript: line 238-240). Secondly, we rewrite Section 2.4 (revised manuscript: line 147-204) and add how we obtain the clock rate into the content as "Bayesian coalescent phylo-dynamic analysis was then performed with BEAST v1.10.4 (Bayesian Evolutionary Analysis Sampling Trees). From the BEAST result, we also obtained the estimation of clock rates. The clock rate here is to estimate the number of substitutions per site per year" (revised manuscript: line 175-178). We hope that this clarifies the confusion.

  8. Page 6, line 168. The method that estimated R0 value in Figure 4b needs to describe in Methods section.

    Response: Thanks! As you and other reviewers suggested, we rewrite Section 2.4 and add the details of the model based on which we estimate R_0. More specifically, you can find it in Section 2.4 about R_0 (revised manuscript: line 185-199), "The reproductive number is calculated according to its definition and estimated from the time-varying effective size. The estimated ratio of reproductive number suggests the transmission advantage of VOC against other variants. We denote the growth rate of hosts carrying non-VOC as r(t) , and that carrying VOC as r'(t) . Both r(t) and r'(t) can be obtained from estimated effective size, and the corresponding time-varying reproductive number of non-VOC and VOC are denoted as and respectively. According to the definition of reproductive number, we can estimate the ratio of reproductive number of VOC against non-VOC as

    $$
    \phi_t^1=\frac{R_t^{'}}{R_t}=\frac{\int_0^{\infty}g(s)e^{-\int_0^s r(t)dt}}{\int_0^{\infty}g(s)e^{-\int_0^s [r(t)+\rho(t)] dt}}
    $$

    Here g(s) is the density of generation times. Utilizing the same token, we can also estimate the ratio through r^{'}(t):

    $$
    \phi_t^2=\frac{R_t^{'}}{R_t}=\frac{\int_0^{\infty}g(s)e^{-\int_0^s [r'(t)-\rho (t)]dt}}{\int_0^{\infty}g(s)e^{-\int_0^s r'(t) dt}}
    $$

    A robust estimator of the ratio can be obtained by averaging the above two estimations, denoted as

    $$
    \widehat{\phi_{t}} = \frac{\phi_t^1+\phi_t^2}{2}
    $$
  9. Page 6, line 170. In addition to the global ratio in Figure 5a, add the curves for the studied countries. Also, add the global value in Figure 5b and 5c.

    Response: Thanks for your suggestion. In Figure 5 in the revised manuscript (line 254-257), we add the global curve to the trend of each country for better comparison, and also mark each country in different colors in the panels.

  10. Page 7, Figure 5c is blank.

    Response: Thank you for your comment. We adjust the figures in the revised manuscript, and hope that it works fine now. Figure 5b (the previous Figure 5c) displays the time-varying ratio of reproductive numbers in each country marked in the colors consistent with Figure 5a.

  11. Page 8, line 191. Why only three countries were selected as analyzing AY.4 variants?

    Reponse: We conducted the analysis on AY.4 to illustrate that our model can also compare the lineages within a variant family, in this case, the Delta variant. However, the advantage of AY.4 over B.1.617.2 is quite slight, and to eliminate the confusion, we keep only the results of AY.4 in the UK in the revised manuscript. The UK is selected due to its high data quality and also the real proportion of AY.4 in Figure 2 matches our prediction. The data quality can be evaluated based on the "index" column in Table 1, where the index of UK is over 0.1, i.e., other countries collected and uploaded <10% of all sequences reported to JHU CSSE. The revision about this in the manuscript can be found in Section 3.3, including both the content and also the figures.

  12. Page 10, line 212. Please indicate all the estimated values of the effects of all 13 NPIs and vaccination in an additional table.

    Response: Thank you for your suggestion. We agree with you that it would be more clear to present the estimated values in a separate table. We add the estimated values and also the 95% confidence intervals in Table 2 in Section 3.5 of the revised manuscript (revised manuscript: line 305-306):

    Table 2. Estimation of effects of 13 NPIs and vaccination

    Intervention Effective size 95% CI
    Covid 19 testing policy -0.64 (-1.09, -0.19)
    Covid contact tracing 0.04 (-0.3, 0.39)
    Covid vaccination policy 0.01 (-0.17, 0.19)
    Debt relief -0.07 (-0.28, 0.14)
    Face covering policies 0.42 (0.05, 0.78)
    Income support -0.09 (-0.48, 0.29)
    International travel -1.68 (-2.32, -1.03)
    Public campaigns 2.24 (1.07, 3.4)
    Public events 0.26 (-0.1, 0.63)
    Public gathering rules 0.32 (-0.13, 0.76)
    Public transport -0.2 (-0.5, 0.11)
    School closures 0.05 (-0.25, 0.35)
    Stay at home -0.13 (-0.46, 0.2)
    Workplace closures -0.32 (-0.69, 0.05)
  13. Page 11, Figure 10. Please display the Delta variants data in UK to make sure the consistent display of all variants in this Figure. The method to predict the spread trend was needed to describe in Method section.

    Response: Thanks for your suggestion. We initially choose the Alpha in the UK and the Delta in the India because they are where the first sequence was collected, and the data in South Africa is too limited thus we select the Omicron in the UK where UK is the first European country that has an outbreak of Omicron. However, we believe that your suggestion provides a wiser solution by simply using the data in the UK for three variants to make it consistent and also eliminate the effect brought by intrinsic fluctuation factors among countries. Thus in the revised manuscript, Figure 9 (the previous Figure 10) displays the estimated spread trend of variants in the UK based on our model, and as the data quality of the UK in much better than of India, we can find that the prediction curve of Delta matches more with the fact.

    In the Method section (Section 2.4), we also add more details about how we predict the spread trend as "Besides, we conducted simulation based on the same statistical model and estimated the date when a VOC becomes dominant (proportion >=50%), by first estimating the posterior distribution of parameters given in the logistic linear model under a Bayesian framework and then simulating the growth curve of the proportion of VOC for 10,000 independent trails" in Section 2.4 (revised manuscript: line 200-204).

  14. Page 11, line 268-270. The description needs reference citations.

    Response: Thanks for your suggestions. As you suggested, we search for the literatures about the comparison of AY.4 and B.1.617.2 and add the following five references in the revised manuscript. Specifically, we found a paper about an Italian male infected by AY.4 variant which, to some extent, supports our conclusion as well from the biological view as well.

    • Celik I, Tallei T E. A computational comparative analysis of the binding mechanism of molnupiravir’s active metabolite to RND-dependent RNA polymerase of wild-type and Delta subvariant AY.4 of SARS-CoV-2. J Cell Biochem 2022, 1-12.

    • Mazur-Panasiuk N, Rabalski L, Gromowski T, et al. Expansion of a SARS-CoV-2 Delta variant with an 872 nt deletion encompassing ORF7a, ORF7band ORF8, Poland, July to August 2021. Eurosurveillance 2021, 26(39), 2100902.

    • Umar M, Ikram A, Rehman Z, et al. Genomic diversity of SARS-CoV-2 in Pakistan during fourth wave of pandemic. medRxiv 2021, 2021.09.30.21264343.

    • Alkhatib M, Bellocchi M, Marchegiani G, et al. First case of a COVID-19 patient infected by Delta AY.4 with a rare deletion leading to a N gene target failure by a specific real time PCR assay: novel Omicron VOC might be doing similar scenario? Microorganisms 2022, 10(2), 268.

    • Singanayagam A, Hakki S, Dunning J, et al. Community transmission and viral load kinetics of the SARS-CoV-2 delta (B.1.617.2) variant in vaccinated and unvaccinated individuals in the UK: a prospective, longitudinal, cohort study. Lancet Infec Dis 2022, 22(2), 183-195.

  15. Page 11, line 270-271. The difference of clock rates in Figure7 and Figure 4a were not obvious.

    Response: Thank you for your comment. As we respond in #7, we clarify the definition of clock rate in the revised manuscript, and also delete part of irrelevant results. We actually wanted to illustrate that the comparison of clock rates between VOC and pre-dominant variant, instead of the difference in the absolute values. We also revised the sentence here into "Based on the results of clock rate, the advantage of B.1.617.2 to its pre-dominant variants is more obvious compared to that within Delta variants. The clock rate of the VOC is also found to be higher than the pre-dominant variant in Figure 4a and Figure 7a" (revised manuscript: line 377-380).

  16. Page 12, line 291-292. The description needs reference citations.

    Response: Thanks for your comment. We search for other literatures about the effect of public health measurements and add three references about this. We also found a paper that supports our conclusions and add it in the manuscript that "Similar research also suggests that the existing public health measures, including mass testing strategies, were effective but insufficient to control the COVID-19 outbreak caused by the Delta variant in Vietnam [20]" (revised manuscript: line 403-405).

  17. Page 13, line 326. Please indicate which result of Russia had limitation and describe the reason that cause limitation of the data from Russia.

    Response: The result of Russia had limitations mainly due to the low number of sequences found on GISAID. To clarify this, in the Discussion Section, we add "The reason of this insufficiency in Russia is mainly due to the lack of sequences on the public database (GISAID), as shown in Table 1, it is estimated to have over 1 million records between February and July in 2021, but only about 1000 complete sequences are uploaded to GISAID. Therefore, the conclusions for countries with low index may not reflect the reality" (revised manuscript: line 440-444).

Reviewer 2 Report

Summary of the Work

The authors proposed a realistic model able to determine if the measures of current policies and vaccination campaign against the SARS-CoV-2 Variant of Concern (VOC) and predominant variants are appropriate and reasonable. More specifically, the aim of this study is to estimate transmissibility of VOC when it appears, using data collected from early stage, before taking any decision such as cutting down any transportation from a region to the rest of the world. To this end, the authors developed a modified version of Bayesian model on the growth rate of a variant from country-level where supplementary factors, such as the interactions between countries and common human interventions - including - vaccination and public transportation policies, are also taken into account. Based on data collected from the early spread of variants, they estimated the additive difference in growth rates and the ratio of reproductive number of VOC to its pre-dominant variant.

Main obtained results

The model is able to estimate the trend of VOC compared to its pre-dominant variant. More specifically, the model can predict when a VOC will become dominant, given the data of its early-stage spread with applications on Alpha, Delta and Omicron lineages.

Minor

M1) Please, specify the acronyms when they appear for the first time in the manuscript even when they are well known in the literature. Please check that all the acronyms introduced in the manuscript are duly specified (e.g., the acronym CI cited in the Abstract is not specified).

M2) Figures 5.(c) and 8.(c) have to be integrated in a more suitable way.

Remarks (General)

R1) The Statistical analysis adopted by the authors it is only vaguely described in the text. The authors stated that “Details of the model can be found in Supplementary Information”. Fine, but will this Supplementary Information be published together with the manuscript? Anyhow, written in this form, the subsection 2.4. “Statistical Analysis” is not attractive and certainly not suitable for a politician who - at least according to what the authors have stated - should be one of the major interlocutors of this study.

R2) There are several points in this study that should be clarified (e.g., the crucial role of the intrinsic fluctuations dynamics especially in the initial phase, the role of the different compartments which, interconnected among them, determine the dynamics of the spreading of the infection, etc.).

R3) The work should be better framed within the studies conducted by scientists proposing models on this topic (see tips 1-3). It is advisable to add some appropriate references about it.

The following suggestions aim to clarify some aspects.

Suggestions

1) The authors proposed a statistical analysis, based on trend from early stage of spread, able provide direct evidence on the transmissibility of VOC and effect of policies. However, this analysis is introduced in the text in a very (too) fleeting way.

Basically, the authors' statistical analysis is described by declaring the following steps:

i) The additive difference of growth rate between VOC and pre-dominant was estimated on the base of an improved Bayesian framework by considering both the correlation among countries;

ii) The Bayesian coalescent phylodynamic analysis was performed with BEAST 131 v1.10.4.;

iii) The estimation of clock rates has been obtained from the BEAST result;

iv) The phylogenetic tree was constructed by maximum likelihood method in IQtree v2.1.3 assuming 133 a HKY model for substitution.

The authors will certainly agree with the reviewer that, written in this way, subsection 2.4. it is not easy to read and the reader, especially if he/she belongs to a purely medical-hospital scientific community or is even a politician, risks getting lost, by ignoring this study consequently. To avoid this, the authors it is suggested to rewrite subsection 2.4. explaining, step-by-step, their procedure (for example, "Why should a Bayesian coalescent phylodynamic analysis be performed?" or, "why is it necessary to estimate clock rates?", etc.) The need to introduce these basic concepts must be clear in the reader's mind. 

2) The model proposed by the authors aims to estimate the transmissibility of VOCs when it appears, using the data collected from early stage. However, as known data from the initial phase are inevitably subjected to large intrinsic fluctuations. Please describe how the authors' model takes into account the effects of fluctuation dynamics.

3) We may object that, scientists working on modelling of SARS-CoV-2 infection, use data collected from early stage as initial conditions for equations governing the dynamics of the various compartments (i.e., susceptible individuals, infectious people etc.). The restrictive measures adopted by the various governments, such as cutting down transportation, isolation of people etc., are subsequently introduced on the basis of these predictions. In this study, however, the initial data are directly used to obtain global trends able to indicate the surveillance strategies that governments should adopt. Please, dispel this possible objection.

4) Figures 3. show the additive difference in growth rate of B.1.617.2 vs collection data and countries. These values are (inevitably) subjected to fluctuations which have not been reported in these estimations. We may object that the effect of these fluctuations is to "cover" the trends shown by these figures. Please, clarify this aspect.

5) Concerning the data reported in Figure 4, please, specify the accuracy and precision of the estimated values for the clock rate and R0 for the variants B.1.617.2 pre-dominant, related to the five countries examined.

6) Please, clarify Figures 5.(c). In Fig. 5.(b), please, specify the accuracy of the estimated of the ratio of reproductive number for the countries.

7) Same suggestion as the point 6) above. Please, integrate Figs 8.(c) in a more suitable way and specify the error related to the estimation of the ratio of reproductive number (against collection data and countries).

Conclusions

I enjoyed reading the study. However, there are some points that in my opinion need to be clarified. Firstly, since this work proposes a model, it is necessary to also take into account how other scientific communities are dealing with the same problem (see questions 1-3 above). The suggestions above aim to highlight some vulnerable aspects of this study and to fill some gaps. In my opinion, answering the above objections will help improve the solidity of the work significantly.

Author Response

Firstly we would like to thank you for reviewing our manuscript, and we benefit a lot from the revising process based on your suggestions. Please see below for our responses and when the line number is mentioned, it refers to the revised manuscript version with marked notes (not the clean revised version).

M1) Please, specify the acronyms when they appear for the first time in the manuscript even when they are well known in the literature. Please check that all the acronyms introduced in the manuscript are duly specified (e.g., the acronym CI cited in the Abstract is not specified).

Response: Thank you for the suggestion! We are sorry that we use the abbreviations in carelessness in the initial submission, and in the revised manuscript, we have added the full names of the abbreviations when they first appear. We believe hope that this can help readers understand the content better.

M2) Figures 5.(c) and 8.(c) have to be integrated in a more suitable way.

Response: Thanks for pointing this out. Regarding to Figure 5a in the revised version, we have added the global trend as the dotted line in each country, and plot them in different colors consistent with Figure 5a. For Figure 7 (the previous Figure 8c), as the analysis is conducted on AY.4 in a similar way as B.1.617.2, we only leave the results of the UK in Figure 7, and therefore the figures become more clear. We hope that this is more suitable compared to the previous version.

Remarks (General)

R1) The Statistical analysis adopted by the authors it is only vaguely described in the text. The authors stated that “Details of the model can be found in Supplementary Information”. Fine, but will this Supplementary Information be published together with the manuscript? Anyhow, written in this form, the subsection 2.4. “Statistical Analysis” is not attractive and certainly not suitable for a politician who - at least according to what the authors have stated - should be one of the major interlocutors of this study.

Response: Thanks for your remark. Initially we do plan to publish the manuscript and the Supplementary Information together, thus we put all mathematical details in the SI in the first submission. As you and another reviewer suggested, we have now moved most details back to the content of manuscript, so that the readers are more aware of how we conduct the analysis and understand the model better. Please find the Section 2.4 in the revised manuscript and also below, with the title "Bayesian model and phylo-dynamics analysis" (line 147-204):

"We denote the count of VOC genomes as B_{i,w} , out of a total count of selected viral sequences T_{i,w} for week i and country w. B_{i,w} follows a binomial distribution with a time-varying probability for each country, where the probability is modelled through a logistic linear model:

$$
B_{i,w} \sim Binom(T_{i,w},p_{i,w})
$$

We assume the log odds of sampling frequency for VOC \phi_{i,w} is controlled by a Markov Chain, where

$$
\phi_{i,w+1} = \phi_{i,w}+\rho_{i,w}
$$

The variable \rho_{i,w} represents the weekly growth rate in frequency of VOC, suggesting the additive difference in growth rate for VOC compared with other lineages. We model \rho_{i,w} from a multivariate normal distribution with non-identity variance-covariance matrix \Sigma . The variance and covariance are estimated through the international travelling inflows and outflows collected from Statista (https://www.statista.com). The principal diagonal entries are scaled to 0.1 as the research by Volz and colleagues [2]. Assume the total number of countries are denoted as K ,

$$
\rho_{w} = (\rho_{1,w},\rho_{2,w},...,\rho_{K,w},) \\ \rho_{w} \sim N(\mu_W,\Sigma)
$$

The 13 NPIs and vaccinations are represented as covariates cov_{i,w} functioning on the average additive difference in growth rate \mu_{i,w} ,

$$
\mu_{w+1} \sim N(\mu_{i,w}+\beta^t cov_{i,w},\sigma_t)
$$

The priors are modelled given similarly:

$$
\phi_{i,1} \sim N(-7,10) \\ \sigma_t \sim Exp(100) \\ \beta \sim N({\bf{0}}_p,{\bf{I}}_p)
$$

Bayesian coalescent phylo-dynamic analysis was then performed with BEAST v1.10.4 (Bayesian Evolutionary Analysis Sampling Trees) [12]. From the BEAST result, we also obtained the estimation of clock rates. The clock rate here is to estimate the number of substitutions per site per year. The phylogenetic tree was constructed by maximum likelihood method in IQtree v2.1.3 [13] assuming a HKY model (Hasegawa-Kishono-Yano model) for substitution. The selection of HKY model is based on the fact that the substitution rates of transitions are higher than expected by chance relative to those of transversions in SARS-CoV-2, which therefore makes the Jukes-Cantor model incompatible in this case [14,15]. Based on the estimated additive difference in growth rate, clock rate for each variant, the ratio of reproductive number was calculated using “mlesky” and “treedater” packages in R software, v4.1.2. The reproductive number is calculated according to its definition and estimated from the time-varying effective size. The estimated ratio of reproductive number suggests the transmission advantage of VOC against other variants. We denote the growth rate of hosts carrying non-VOC as r(t) , and that carrying VOC as r'(t) . Both r(t) and r'(t) can be obtained from estimated effective size, and the corresponding time-varying reproductive number of non-VOC and VOC are denoted as and respectively. According to the definition of reproductive number, we can estimate the ratio of reproductive number of VOC against non-VOC as

$$
\phi_t^1=\frac{R_t^{'}}{R_t}=\frac{\int_0^{\infty}g(s)e^{-\int_0^s r(t)dt}}{\int_0^{\infty}g(s)e^{-\int_0^s [r(t)+\rho(t)] dt}}
$$

Here g(s) is the density of generation times. Utilizing the same token, we can also estimate the ratio through r^{'}(t):

$$
\phi_t^2=\frac{R_t^{'}}{R_t}=\frac{\int_0^{\infty}g(s)e^{-\int_0^s [r'(t)-\rho (t)]dt}}{\int_0^{\infty}g(s)e^{-\int_0^s r'(t) dt}}
$$

A robust estimator of the ratio can be obtained by averaging the above two estimations, denoted as

$$
\widehat{\phi_{t}} = \frac{\phi_t^1+\phi_t^2}{2}
$$

R2) There are several points in this study that should be clarified (e.g., the crucial role of the intrinsic fluctuations dynamics especially in the initial phase, the role of the different compartments which, interconnected among them, determine the dynamics of the spreading of the infection, etc.).

Response: Thanks for your suggestion! We studied the intrinsic fluctuations dynamics in epidemiology literatures, and we agree with you that it is an important issue that we should consider. For the analysis of B.1.617.2, the initial phase should be its appearance to the quick spread, i.e., months since its appearance in India in late 2020. We select the time period as 2021-02-21 to 2021-07-17, mainly because that the initial stage is noisy, as you pointed out. Thus our model avoids this issue by ignoring the initial stage. To provide more proof of this, in the analysis of prediction the date of VOC becoming dominate, we found out that other variants, such as Beta, Gamma, and Mu, all fail to reach the 1% of daily collected sequences, therefore they didn't become a global threat such as Delta and Omicron. This indicates that we may be safe to say that, a variant should be of our concern if its proportion is over 1% in the daily records on GISAID. Our model is focusing the period after a VOC reaches 1% in the daily records, and is able to compare VOC to its pre-dominant variant in this sense. Since our model avoids the very initial time period, the compartments that determine the dynamics of the spreading of the infection are also out of our consideration.

Meanwhile, as you pointed out, we also consider the different factors after the initial stage to see if they have an impact on our model and the conclusions. Firstly, the noise is eliminated by the smoothing technology (averaging to weeks instead of daily records, except for Omicron variant, because of its super-high transmissibility), and therefore the curves in the figures don't fluctuate rapidly. Secondly, our model focuses on the population of cases, instead of the whole population, therefore the variables in the traditional SIR models make little difference on the infectious population. We agree to your statement that the susceptible individuals and infectious people have an impact on the spread of variant, while the statistics data of daily records on GISAID already covers the influence. And thirdly, the effect of some compartments are implicitly included in the model. For example, the different scales of population in each country can be inferred based on the variable T_{i,w}, and the 13 NPIs and vaccination variables are also included in our model. The vaccinated population, in India, Indonesia and Russia takes no more than 20% of the whole country population until August 2021, and the ratio is about 50% for the US and the UK. Therefore, most of the population in these countries remain susceptible during the studied time period, February to July 2021. However, it is worthy to add this in the Discussion Section, because now the susceptible population is much smaller since 2022. We add it in the Discussion Section that "Finally, our model doesn’t consider the intrinsic fluctuations dynamics and the compartments determining the dynamics. Our model avoids the fluctuation issue in the initial stage by avoiding the days after the very first appearance of VOC, however, the prediction model can be improved substantially if it can detect a VOC since its appearance. Besides, the factors affecting the spread of SARS-CoV-2 [26-28], such as the vaccination rate, should be taken into consideration as in some countries a large proportion of people have been fully vaccinated since 2022, leading to an entirely different environment compared to early 2021" in the revised manuscript (line 449-456).

R3) The work should be better framed within the studies conducted by scientists proposing models on this topic (see tips 1-3). It is advisable to add some appropriate references about it.

Response: The model is mainly an improvement of the work by Volz et al. (reference 2 in the manuscript), but as you suggested, more references help solve certain issues and clarify some aspects. We added the research conducted by other scientists proposing models using SIR and evolutionary analysis of the dynamics in the Discussion Section as "Other epidemiology research may apply the SIR (susceptible-infected-recovered) and SEIR (susceptible-exposed-infected-recovered) models for this problem, and a possible way might be combining our Bayesian model with the SIR model to solve this issue [29-31]" (line 456-459) in the revised manuscript.

New references:

  • Calafiore G C, Novara C, Possieri C. A modified SIR model for the COVID-19 contagion in Italy. 2020 59th IEEE Conference on Decision and Control (CDC) 2020, 3889-3894.

  • Ellison G. Implications of heterogeneous SIR models for analyses of COVID-19. NBER working papers series 2020,118, 27373.

  • He S, Peng Y, Sun K. SEIR modeling of the COVID-19 and its dynamics. Nonlinear Dyn 2020, 101, 1667-1680.

Suggestions

1) The authors proposed a statistical analysis, based on trend from early stage of spread, able provide direct evidence on the transmissibility of VOC and effect of policies. However, this analysis is introduced in the text in a very (too) fleeting way.

Basically, the authors' statistical analysis is described by declaring the following steps:

i) The additive difference of growth rate between VOC and pre-dominant was estimated on the base of an improved Bayesian framework by considering both the correlation among countries;

ii) The Bayesian coalescent phylodynamic analysis was performed with BEAST 131 v1.10.4.;

iii) The estimation of clock rates has been obtained from the BEAST result;

iv) The phylogenetic tree was constructed by maximum likelihood method in IQtree v2.1.3 assuming 133 a HKY model for substitution.

The authors will certainly agree with the reviewer that, written in this way, subsection 2.4. It is not easy to read and the reader, especially if he/she belongs to a purely medical-hospital scientific community or is even a politician, risks getting lost, by ignoring this study consequently. To avoid this, the authors it is suggested to rewrite subsection 2.4. explaining, step-by-step, their procedure (for example, "Why should a Bayesian coalescent phylodynamic analysis be performed?" or, "why is it necessary to estimate clock rates?", etc.) The need to introduce these basic concepts must be clear in the reader's mind.

Response: Thanks for your suggestion. To better illustrate our model, we have re-written Section 2.4 as mentioned above during the revision, especially focusing on the main framework of the model that we proposed. We also clarify some terminology, such as the clock rate in Section 2.4 (revised manuscript: line 177-178) and Section 3.2 (revised manuscript: line 238-240) in the manuscript, so that readers from medical and biological background can understand our model and the results better.

2) The model proposed by the authors aims to estimate the transmissibility of VOCs when it appears, using the data collected from early stage. However, as known data from the initial phase are inevitably subjected to large intrinsic fluctuations. Please describe how the authors' model takes into account the effects of fluctuation dynamics.

Response: It is right that we try to estimate the transmissibility of VOCs during the early stage, but it is necessary to point it out that we don't monitor all new variants once they emerge. Based on history experience, the other variants, including the local popular ones such as Beta and Mu, don't even reach to 1% of the daily collected records on GISAID. Thus for the Delta variants, we analyzed the period since February 2021, rather than late 2020. This actually proves the reviewer's correctness in this point, because it is extremely difficult to analyze the data when only few sequences are collected and to predict its transmissibility in epidemiological view. Possible ways include biological experiments on artificial sequences on animals, but that would be out of the topic of this paper. However, as we answered in R2) above, we have added this in Section 3.6 (revised manuscript: line 308-314) and also in the Discussion Section, and hope to improve the model in future to handle this issue.

3) We may object that, scientists working on modelling of SARS-CoV-2 infection, use data collected from early stage as initial conditions for equations governing the dynamics of the various compartments (i.e., susceptible individuals, infectious people etc.). The restrictive measures adopted by the various governments, such as cutting down transportation, isolation of people etc., are subsequently introduced on the basis of these predictions. In this study, however, the initial data are directly used to obtain global trends able to indicate the surveillance strategies that governments should adopt. Please, dispel this possible objection.

Response: We totally agree with you that the government make policies after the initial stage to face the new VOC, and in our model, we treat the policies as the covariates, similar in the natural way, i.e., when the pandemic happens, there exist the current policies against the pre-dominant variant, and during the spread of VOC, the policy may or may not change based on the prediction of whether this new variant will become a real threat. And we suggest the government to emphasize the policy on the controlling public information campaigns and advocating facial coverings, as they are found to have lower effect on the Delta variant compared to the pre-dominant variants. This result is helpful for the government to determine the policy against, for example, Omicron variants, as we already have evidence that even Delta can escape from these two aspects compared to the older variants. There will be no direct proof to illustrate the effect of policies on Omicron variants until we get the data collected during Omicron's expansion, but the history we learned from Delta can help us prepare better for the new variants. To clarify this, we add "Please note that the suggestion is based on our history about Delta, rather than the direct evidence on the Omicron variant" in the Discussion Section (revised manuscript: line 414-415).

4) Figures 3. show the additive difference in growth rate of B.1.617.2 vs collection data and countries. These values are (inevitably) subjected to fluctuations which have not been reported in these estimations. We may object that the effect of these fluctuations is to "cover" the trends shown by these figures. Please, clarify this aspect.

Response: Regarding to the issue of fluctuations, please refer to our response to R2).

5) Concerning the data reported in Figure 4, please, specify the accuracy and precision of the estimated values for the clock rate and R0 for the variants B.1.617.2 pre-dominant, related to the five countries examined.

Response:The accuracy and precision are not applicable in this case, because 1) the clock rate is continuous rather than a binary label 2) the real clock rate is unknown to us thus there is no gold standard to verify the results, and all we can do is to estimate the clock rate by various approaches.

6) Please, clarify Figures 5.(c). In Fig. 5.(b), please, specify the accuracy of the estimated of the ratio of reproductive number for the countries.

Response: Thanks for your suggestion. We have added the global result in Figure 3 and Figure 5 to clarify the results. Regarding to the ratio reproductive, please refer to our response to 5).

7) Same suggestion as the point 6) above. Please, integrate Figs 8.(c) in a more suitable way and specify the error related to the estimation of the ratio of reproductive number (against collection data and countries).

Response: We have modified the results about AY.4 to limit the results to the United Kingdom to make the point clear, and also re-arranged Figure 7 (the previous Figure 8). As the R_0 is always continuous, unknown, and can only be estimated, the error would be inapplicable here.

Conclusions

I enjoyed reading the study. However, there are some points that in my opinion need to be clarified. Firstly, since this work proposes a model, it is necessary to also take into account how other scientific communities are dealing with the same problem (see questions 1-3 above). The suggestions above aim to highlight some vulnerable aspects of this study and to fill some gaps. In my opinion, answering the above objections will help improve the solidity of the work significantly.

Response: We would like to sincerely thank you for the helpful comments that you have on our manuscript, and we admit that the manuscript didn't consider certain details that you pointed out. During the revision, our understanding about both our model and its applications improves and also realizes its limitations, which we benefit a lot from your review, and surely will be our future research interest.

Reviewer 3 Report

This submission presents analysis methods that it claims offer an improvement over those of Volz et al regarding why and how one SARS-CoV-2 VOC strain is displacing another and when it will become dominant. The proposed improvements utilize additional information in the analyses performed, such as the correlation among countries and the effect of NPIs and vaccinations. The analyses presented use GSAID sequences and daily case reports from Johns Hopkins University (JHU). The logic presented assumes GSAID submission is independent of variant information and that JHU daily report is accurate. Both assumptions are more problematic than any of the assumptions used in Volz et al. The analyses were performed across 5 countries with different histories of variants before Omicron. There are interesting differences between these 5 countries with regard to how the Delta variant rose to dominance. But the results of those differences are only very superficially analyzed. For example, the delta substitution rates in Indonesia are considerably above those of the other countries. But how this affects the major predictions made is not discussed adequately. Likewise the early dynamics in India are quite different from the other countries examined. Similarly the clock rate differences  between AY.4 and B.1.617.2 are markedly different between India, UK, and USA. What is going on there and how that might affect the inferences made is inadequately discussed. Thus, while the differences between the five countries are quite notable, insufficient work was performed to extract meaningful inferences as to what these differences represent.

There is a lack of focus on either methods used or new understanding generated by the analyses performed. Likewise there is a lack of depth in explaining the results observed. It is possible that, in part, these deficiencies derive from linguistic differences between Chinese and English. But the first step in overcoming these deficiencies might be for the paper to be clear what its major message is. Is this a methods paper or a new knowledge generated paper? More papers using the source data this paper used could help to better understand what happened and why it happened. Maybe it would be enough for this paper to present the differences between the five countries examined. Whatever path is chosen for the paper, tightening up the description of the findings, or more clearly explaining why and how the methods used uncovered the new knowledge that is presented, would give the paper a more understandable focus.

The estimation of the domination date does not impress me as having either meaningful theory or infection control utility.

The analyses presented are interesting. But a major effort to refocus the work to make more clearly useful theoretical or practical inferences is needed. It is possible that I might have missed some value in this paper because the English used was hard to understand. If there is a more experienced English speaker who could be involved in writing the paper, that might help.

Figures 5c and 8c are blank. They are said to present time trends.

Author Response

Firstly we would like to thank you for reviewing our manuscript, and we benefit a lot from the revising process based on your suggestions. Please see below for our responses and when the line number is mentioned, it refers to the revised manuscript version with marked notes (not the clean revised version).

Review and responses:

This submission presents analysis methods that it claims offer an improvement over those of Volz et al regarding why and how one SARS-CoV-2 VOC strain is displacing another and when it will become dominant. The proposed improvements utilize additional information in the analyses performed, such as the correlation among countries and the effect of NPIs and vaccinations. The analyses presented use GSAID sequences and daily case reports from Johns Hopkins University (JHU).

The logic presented assumes GSAID submission is independent of variant information and that JHU daily report is accurate. Both assumptions are more problematic than any of the assumptions used in Volz et al.

Response: Thank you for pointing this out and reading our manuscript carefully. We agree with you that both assumptions seem too theoretical and difficult to realize in practice, and we would like to illustrate that the data source that we use is public and free, in contrast to the data that Volz et al. used in their research is collected by the CDC in UK. When we began this project, we sent an email to Professor. Volz and asked about the data source, but he cannot share it with us due to confidential issues. Besides, our project aims to study the major countries which, requires the accurate data for all major countries. So far, GISAID is the largest public database that we can find and many literatures use the same data source. As we can also tell from Table 1, we define the "index" to evaluate the quality of data, and we are aware of the fact that even for the countries with a high index, it only implies that they collect and submit more sequences to GISAID, but it should be safe to say that the data for a country that has millions of cases but only shares dozens of sequences is unreliable. Therefore in the analysis of AY.4 and Omicron (in the revised manuscript), we only select data from the UK considering its high index. If data with better quality is available, we believe that the model will do a better job, but so far that is really the best we can do. To illustrate this point in the manuscript, we use the data from Russia as an example and in the Discussion Section we add "The reason of this insufficiency in Russia is mainly due to the lack of sequences on the public database (GISAID), as shown in Table 1, it is estimated to have over 1 million records between February and July in 2021, but only about 1000 complete sequences are uploaded to GISAID. Therefore, the conclusions for countries with low index may not reflect the reality" in revised manuscript line 429-433.

The analyses were performed across 5 countries with different histories of variants before Omicron. There are interesting differences between these 5 countries with regard to how the Delta variant rose to dominance. But the results of those differences are only very superficially analyzed. For example, the delta substitution rates in Indonesia are considerably above those of the other countries. But how this affects the major predictions made is not discussed adequately.

Response: Thank you for your suggestion. The difference among five countries, actually, can also be found in Table 1, as the number of records on GISAID are not even on the same scale. The clock rate is defined as the rate of nucleotide changes in units of calendar time, therefore, the B.1.617.2 changes in a higher rate in Indonesia than in other countries. We didn't dig into this issue mainly because that there are only 979 B.1.617.2 sequences on GISAID, and therefore the estimation might not be accurate. The pre-dominant variant in Indonesia is B.1.466.2, which also has mutations in the spike protein such as D614G, N4239K and P681R, while in the meantime, most countries in the world are overwhelmed by the Alpha variants (such as the UK and US in our manuscript). Another reason might be that the appearance of B.1.617.2 in Indonesia is relatively late compared to in other countries in Figure 6, suggesting that a lot of mutations may have already happened in other countries and then entered Indonesia, causing its high clock rate. We believe that those factors may affects the spread of the VOC, but that is out of the scope of our project. In Figure 4, we mainly want to show the contrast between the B.1.617.2 and the pre-dominant variant. To clarify this, we add "The clock rate here is to estimate the number of substitutions per site per year as defined and calculated in BEAST based on HKY model. And the reason for the high clock rate of B.1.617.2 in Indonesia might be related to its late appearance and the different pre-dominant variants" in the revised manuscript (line 238-241).

Likewise the early dynamics in India are quite different from the other countries examined. Similarly the clock rate differences between AY.4 and B.1.617.2 are markedly different between India, UK, and USA. What is going on there and how that might affect the inferences made is inadequately discussed. Thus, while the differences between the five countries are quite notable, insufficient work was performed to extract meaningful inferences as to what these differences represent.

Response: In the revised manuscript, we have removed the previous clock rate of AY.4 and only leave the result of AY.4 in the UK instead of three countries. The main reason is that the analysis procedure is similar to B.1.617.2, and the advantage of AY.4 over B.1.617.2 is quite slight in other countries. However, we fully agree with you that this will affect the inferences. From our perspective, the clock rate reflects how quickly the variant changes, and the corresponding results would be the additive weekly difference and reproductive numbers. Thus we are not analyzing why B.1.617.2 or other lineages outcompete the pre-dominant variants, but estimating how much VOC is more transmissible. To clarify this, we add the definition of clock rate in the revised manuscript (line 238-241) as "The clock rate here is to estimate the number of substitutions per site per year as defined and calculated in BEAST based on HKY model. And the reason for the high clock rate of B.1.617.2 in Indonesia might be related to its late appearance and the different pre-dominant variants".

There is a lack of focus on either methods used or new understanding generated by the analyses performed. Likewise there is a lack of depth in explaining the results observed. It is possible that, in part, these deficiencies derive from linguistic differences between Chinese and English. But the first step in overcoming these deficiencies might be for the paper to be clear what its major message is. Is this a methods paper or a new knowledge generated paper? More papers using the source data this paper used could help to better understand what happened and why it happened. Maybe it would be enough for this paper to present the differences between the five countries examined. Whatever path is chosen for the paper, tightening up the description of the findings, or more clearly explaining why and how the methods used uncovered the new knowledge that is presented, would give the paper a more understandable focus.

Response: First to answer the question, our manuscript aims at proposing the method which can be considered as an important improvement of the project by Volz et al. To emphasize this, we rewrite the Section 2.4 to illustrate the model in a clearer way, with the new title "Bayesian model and phylo-dynamics analysis" instead of the original "Section 2.4 Statistical Analysis":

"We denote the count of VOC genomes as B_{i,w} , out of a total count of selected viral sequences T_{i,w} for week i and country w. B_{i,w} follows a binomial distribution with a time-varying probability for each country, where the probability is modelled through a logistic linear model:

$$
B_{i,w} \sim Binom(T_{i,w},p_{i,w})
$$

We assume the log odds of sampling frequency for VOC \phi_{i,w} is controlled by a Markov Chain, where

$$
\phi_{i,w+1} = \phi_{i,w}+\rho_{i,w}
$$

The variable \rho_{i,w} represents the weekly growth rate in frequency of VOC, suggesting the additive difference in growth rate for VOC compared with other lineages. We model \rho_{i,w} from a multivariate normal distribution with non-identity variance-covariance matrix \Sigma . The variance and covariance are estimated through the international travelling inflows and outflows collected from Statista (https://www.statista.com). The principal diagonal entries are scaled to as the research by Volz and colleagues [2]. Assume the total number of countries are denoted as K ,

$$
\rho_{w} = (\rho_{1,w},\rho_{2,w},...,\rho_{K,w},) \\ \rho_{w} \sim N(\mu_W,\Sigma)
$$

The 13 NPIs and vaccinations are represented as covariates cov_{i,w} functioning on the average additive difference in growth rate \mu_{i,w} ,

$$
\mu_{w+1} \sim N(\mu_{i,w}+\beta^t cov_{i,w},\sigma_t)
$$

The priors are modelled given similarly:

$$
\phi_{i,1} \sim N(-7,10) \\ \sigma_t \sim Exp(100) \\ \beta \sim N({\bf{0}}_p,{\bf{I}}_p)
$$

Bayesian coalescent phylo-dynamic analysis was then performed with BEAST v1.10.4 (Bayesian Evolutionary Analysis Sampling Trees) [12]. From the BEAST result, we also obtained the estimation of clock rates. The clock rate here is to estimate the number of substitutions per site per year. The phylogenetic tree was constructed by maximum likelihood method in IQtree v2.1.3 [13] assuming a HKY model (Hasegawa-Kishono-Yano model) for substitution. The selection of HKY model is based on the fact that the substitution rates of transitions are higher than expected by chance relative to those of transversions in SARS-CoV-2, which therefore makes the Jukes-Cantor model incompatible in this case [14,15]. Based on the estimated additive difference in growth rate, clock rate for each variant, the ratio of reproductive number was calculated using “mlesky” and “treedater” packages in R software, v4.1.2. The reproductive number is calculated according to its definition and estimated from the time-varying effective size. The estimated ratio of reproductive number suggests the transmission advantage of VOC against other variants. We denote the growth rate of hosts carrying non-VOC as r(t) , and that carrying VOC as r'(t) . Both r(t) and r'(t) can be obtained from estimated effective size, and the corresponding time-varying reproductive number of non-VOC and VOC are denoted as and respectively. According to the definition of reproductive number, we can estimate the ratio of reproductive number of VOC against non-VOC as

$$
\phi_t^1=\frac{R_t^{'}}{R_t}=\frac{\int_0^{\infty}g(s)e^{-\int_0^s r(t)dt}}{\int_0^{\infty}g(s)e^{-\int_0^s [r(t)+\rho(t)] dt}}
$$

Here g(s) is the density of generation times. Utilizing the same token, we can also estimate the ratio through r^{'}(t):

$$
\phi_t^2=\frac{R_t^{'}}{R_t}=\frac{\int_0^{\infty}g(s)e^{-\int_0^s [r'(t)-\rho (t)]dt}}{\int_0^{\infty}g(s)e^{-\int_0^s r'(t) dt}}
$$

A robust estimator of the ratio can be obtained by averaging the above two estimations, denoted as

$$
\widehat{\phi_{t}} = \frac{\phi_t^1+\phi_t^2}{2}
$$

According to your another suggestion, we have also added the differences between the five countries in Section 2.2 as "However, the difference among the five countries, including the population scale, various pre-dominants, vaccination level, may affect the inferences. To solve this issue, we conduct the analysis on AY.4 based on the data from UK, and the results coincide with the fact that AY.4 outcompetes B.1.617.2 in the UK, as shown in Figure 2b" in the revised manuscript (line 134-138).

The estimation of the domination date does not impress me as having either meaningful theory or infection control utility.

Response: During our analysis, we found that other variants, including the local popular ones such as Beta and Mu, don't even reach to 1% of the daily collected records on GISAID. And a possible strategy for the government could be that, due to the high cost to monitor on all thousands of variants everyday, it pays special attention to the variants that take up 1% of the daily collection on GISAID, and then our model can be used as a good prediction of when this variant will become dominant under the current anti-pandemic policies. We add this in Section 3.6 of the revised manuscript (line 308-314): "In order to estimate if and when a VOC becomes dominating, we found that most variants such as Beta and Mu fail to reach to 1% of the daily collection on GISAID. On the other hand, Alpha, Delta and Omicron all reach to almost 100% in the end, though in different rates. Therefore, a possible strategy for the government could be focusing on the VOCs that take up over 1%, and then conduct analysis on the available data and predict when the VOCs become dominant (>50%). As the results in Figure 9 suggests, our prediction of Alpha, Delta and Omicron is similar to the real date that VOC becomes dominate", and hope that it helps readers understand the utility better.

Please also note that according to another reviewer's suggestion, in the prediction of domination date, we conduct the analysis now all in the UK rather than the countries where the first VOC was collected. The new results are updated in Section 3.6, and also the new Figure 9 (the previous Figure 10). It matches better for Delta in the UK compared to India, partly due to the higher data quality.

The analyses presented are interesting. But a major effort to refocus the work to make more clearly useful theoretical or practical inferences is needed. It is possible that I might have missed some value in this paper because the English used was hard to understand. If there is a more experienced English speaker who could be involved in writing the paper, that might help.

Response: Thanks you for reviewing our manuscript and we learn a lot from your comments. We have read through the manuscript and corrected the typos and grammar errors, including the sentences that may cause confusion. We believe that the revised manuscript is substantially improved with the help of your review.

Figures 5c and 8c are blank. They are said to present time trends.

Response: Thank you for your comment. We adjust the figures in the revised manuscript, and hope that it works fine now. Figure 5b (the previous Figure 5c) displays the time-varying ratio of reproductive numbers in each country marked in the colors consistent with Figure 5a. In Figure 7 (the previous Figure 8), to simplify, we keep only the results globally and in the UK. We think it is also necessary to point out that conducting analysis on the global level is not very informative because different countries have different policies against the pandemic on the same days. Thus the global level here is actually an average of the countries, rather than consider all countries together as a whole.

Round 2

Reviewer 1 Report

This article could be accept in present form.

Author Response

We would like to thank you for your help on our manuscript.

Reviewer 2 Report

The authors have satisfactorily answered all the questions raised in my previous report. My main concern was that the statistical analysis performed by the authors was not described in a clear way. As a consequence, the work risked not being attractive enough and not suitable at all for a politician, who is supposed to be one of the major interlocutors of this study.
This new version illustrates in a more explicit way the methodology adopted by the authors. The authors also discussed in more detail the role of the intrinsic fluctuations dynamics and the different compartments which are interconnected each with others, which have a big impact on the dynamics of the spreading of the infection. In my opinion, this version of the manuscript deserves to be published.

Author Response

We would like to thank you for your help on our manuscript, and the typos and errors have been checked again in this round of revision.

Reviewer 3 Report

Although I checked the "accept after minor revision" category. The changes needed are fairly major. I addressed different things in my current comments than in my previous review because of the now clearly stated purpose of the paper.

Author Response

We would like to thank you for your careful reading and valuable suggestions. We have revised the manuscript accordingly, and the responses can be found below.

The authors now make it clear that they want this paper reviewed for its methodological contributions rather than for the interesting information on VOC in five countries that they present. The major methodological innovation that they present is a method to predict the dynamics of growth in the early stages of a VOI or VOC. Their method “estimated the numbers of variants based on both complete genome records on GISAID and daily report from the Johns Hopkins University Center for Systems Science and Engineering (JHU CSSE) [9]. Under two assumptions that 1) the submission to GISAID is independent of variant information and 2) JHU daily report is accurate on the total cases, proportion of variants in each country is identical to those uploaded to GISAID and thus the number of variants can be estimated.” They also used publicly available data on NPIs. But they do not present the deficiencies and biases that are intrinsic to these data. For example, they do not assess how sampling for sequences is influenced by prior finding of variants in the different countries.

Response: Thank you for your suggestion. Data quality is a fundamental issue, especially to the data-driven projects. In our manuscript, there are three data sources in this project: GISAID, JHU CSSE, and OxCGRT (the NPI and vaccination data). We analyzed them one by one in the following content:

  • GISAID (https://www.gisaid.org) is a database that promotes the rapid sharing of data from all influenza viruses and the coronavirus causing COVID-19. And according to its website, "all users agree that no restrictions shall be attached to data submitted to GISAID, to promote collaboration among researchers on the basis of open sharing of data and respect for all rights and interests". Therefore, we believe that GISAID provides a reliable source of the complete SARS-CoV-2 genomes. However, due to the different levels of medical systems in different countries, some countries submit millions of complete sequencing results, while other countries only contribute few. That is why we do not use the GISAID data as the total number of cases, simply as to estimate the proportion of each variant. The assumption is that the labs or CDC collects samples before knowing the variant information, then conduct sequencing. The process of uploading to GISAID is independent of the sequencing results, thus the records on GISAID can be considered as a sample from the whole cases set in the population. To illustrate this better in the manuscript, we revise the manuscript accordingly:

    • In "Section 2.2 Estimation of number of variants", add "Considering the fact that all users of GISAID agree that no restrictions shall be attached to data submitted to GISAID, it is fairly safe to accept the first assumption."

  • The list of all sources ever used in the data of JHU CSSE (https://github.com/CSSEGISandData/COVID-19) can be found on its website, mostly collected by authorities administrations, such as national government and CDC. There are three main data sources on the latest COVID-19, World Health Organization (WHO), Coronavirus COVID-19 Global Cases by the Center for Systems Science and Engineering at Johns Hopkins University (JHU CSSE), and the European Centre for Disease Prevention and Control (ECDC). The trends of the three data sources are very close, while the JHU CSSE usually has a higher number, possibly because that the cases that collected from state or local labs are not included in the other two data sources. However, we agree with you that this may cause some issues, thus we revise the manuscript accordingly:

    • In "Section 2.2 Estimation of number of variants", add "Regarding to the second assumption, we select JHU daily reports because it includes the confirmed by both the national labs and state or local labs compared to the data source from WHO and European Center for Disease Control and Prevention (ECDC). However, the trend from JHU CSSE, WHO, and ECDC are very similar [10]."

  • The 13 NPIs and vaccination data is from "Our World in Data", sourced from the Oxford Coronavirus Government Response Tracker (OxCGRT) (https://ourworldindata.org/policy-responses-covid). The tracker presents data collected from public sources by a team of over 100 Oxford University students and staff from every part of the world. The data and charts are regularly updated based on the latest version of the response tracker, and describes variation in government responses. Thus it is a great data source to explore whether the government response affects the rate of the infection and identify correlates of more or less intense responses. We make the following revisions in the manuscript:

    • In "Section 2.3 NPI and vaccinations", add "The resource is from the Oxford Coronavirus Government Response Tracker (OxCGRT), tracking the systematic information on policy measures that governments have taken to tackle COVID-19, covering over 180 countries and coded into indicators. The policies are recorded on a scale to reflect the extent of government action, and scores are aggregated into a suite of policy indices. "

  • We have also added this limitation in the Discussion Section that "It is important to point out that our project is data-driven, thus the reliability of data sources is a fundamental challenge and we haven’t presented the deficiencies and biases that are intrinsic to these data. However, all the three datasets are widely used in literatures about COVID-19 [16-29] and are collected by authorities such as national governments and department of public health. Certain issues of the data can be addressed in our model, for example, the lag in submissions is solved by removing sequences too recent or show an abnormal of decrease. We also conduct a sensitivity analysis on the assumption about JHU CSSE daily case number, and the result shows that our model is robust regarding to the total case number (details in the Supplementary Material). If any better data source is accessible in the future, the results based on our model would be improved substantially too.".

Consistent with the focus on methodology, they now present more of the statistical methodology in the main text. They “assume the log odds of sampling frequency for VOC is controlled by a Markov Chain”. But they do not discuss the reality that sample collection in most countries is not consistent with a Markov chain because the finding of a new variant is often followed by intensive sequencing of related cases and this varies considerably between countries. Neither do they conduct any analyses to see how their predictions could be affected by inconsistencies with their assumptions. But at least they clarify how they estimate the ratio of reproductive numbers across VOC. There is a confusion on my part regarding what they refer to as non-VOC. It seems in most cases they are examining ratios of one VOC to another VOC rather than a non-VOC.

Response:

  • Thank you for pointing this out. It is true that the finding of a new variant is often followed by intensive sequencing of related cases and this varies between countries, however, this is especially important for the countries with "zero-case" policy, such as China and New Zealand. To eliminate the issue this might bring, our model doesn't use one day as the smallest unit but conducts the analysis based on weekly data. In the countries that we consider in this project, for example, in the US, the daily new cases can reach 1.4 million, thus the fluctuation between several days generated from a local outbreak actually plays a very small role in the number of weekly confirmed cases from JHU CSSE. Regarding to GISAID, researchers are blind to the variant information because only positive testing result is available to them before sequencing. Thus the sampling process can be considered random.

  • Regarding to the Markov chain for the log odds of sampling frequency for VOC, we use a random walk prior controls autocorrelation in the VOC frequency and growth rate over time. Apparently, in a specific region, the log odds of proportion of VOC in week w+1 should be positively related to the log odds in week w, plus the weekly growth rate of VOC. If the VOC has a higher transmissibility than the pre-dominant variant, then weekly growth rate of VOC is positive, thus the log odds in week w+1 shall be higher. This is the same in Volz et al. 2021.

  • Regarding to the issue about assumptions, we have responded in #1 above.

  • Regarding to the terminology of VOC and non-VOC, we use VOC, variant of concern, to represent the new emerging variant, and other variants as non-VOC. For example, when B.1.617.2 emerges, we want to compare it with the pre-dominant variants, such as B.1.1.7, thus B.1.1.7 is not our "variant of concern" because we are now concerned about B.1.617.2. It is same when AY.4 emerges in July 2021, that we compare the VOC, AY.4, to the pre-dominate variant, B.1.617.2, which WAS a VOC, but now is only a pre-dominant variant. To clarify this, we revise the manuscript accordingly:

    • In Section 3.1 Global trend and major countries, add " When we considered B.1.617.2 as the VOC, we selected the pre-dominant variant in each country for comparison with VOC. When AY.4 becomes the new VOC, we found that the pre-dominant variant is B.1.617 during 2021-05-30 to 2021-07-24. In all analysis, we compare the VOC with its pre-dominant variant. "

There is also confusion on my part regarding their prediction of the time that the new VOC replaces the old VOC. This is done by simulating the model for which they have estimated parameters. Is this prediction one of the major benefits to come out of their methods? If so, how do their predictions compare to those of other methods involving simulation of VOC dynamics? How sensitive are predictions to inconsistencies of reality with the model assumptions? There should be some analyses that introduce bias and assess how that changes the inferences made.

Response: Regarding to the reason why we want to make this prediction of dominating date, firstly, it shows that our model makes a good prediction to the observed trend for three waves (Alpha, Delta, and Omicron), thus proves it to be a reliable model; secondly, the prediction of dominating time gives the government a quantitive measurement of the transmissibility of the VOC to be easily interpreted, compared to the widely used R_0. R_0 is popular in epidemiology, but to non-epidemiological researchers or public, it is easier to understand how long it takes for the VOC to dominate. This can also help the government design public health policies against VOC, for example, if a VOC such as Omicron is predicted to dominate within one week, most policies, especially the wild ones (suggestions on avoiding public crowd, wearing masks), won't have any effect, compared to the strict ones (such as forbidding public gathering, no entering into buildings without masks). It is also important to point it out that the parameters that we used to make the prediction are all only based on data in early period (1% - 10%), thus the prediction is effective.

We searched in PubMed using keywords "variant, dominate, predict", but didn't find other literatures making such predictions. Therefore the comparison between ours and other methods can only be done from the perspective of methodology. Most VOC dynamics focus on the estimation of R_0, or in a local country, which fail to consider the impact that other countries may cause. There are also many literatures evaluating the effect of policies in a certain country, while our model conducts on the international level.

Regarding to the sensitivity analysis, especially to the assumptions, we conduct two more analysis on the B.1.617.2, when the number of cases are 0.5 or 2 times of the current size, i.e., if JHU CSSE data source has some variations. The results show that our model is robust to the data size, i.e., the data from JHU CSSE, and we add this result in the Discussion Section: "We also conduct a sensitivity analysis on the assumption about JHU CSSE daily case number, and the result shows that our model is robust regarding to the total case number (details in the Supplementary Material).". The details of this part is added as a new section "Sensitivity Analysis" in the Supplementary Material. Regarding to the assumption about proportion of VOC and pre-dominant variant, because our model relies on the early expansion of VOC to predict the following trend, any disturbance to the VOC will affect the result and the corresponding conclusions are meaningless to us, the proportion remains same as the input data.

In figure 6 there are no red dots indicating the date of the first B.1.617.2 sequence.

Response: We believe that in the revised manuscript you can find the red dots in Figure 6.

I do not think that the effect sizes in table 2 make a useful contribution to this paper.

Response: We add Table 2 in the last round of revised manuscript according to another reviewer's suggestion. Though most 95% confidence intervals cover 0 and seem insignificant, it is still our responsibility to present the results and draw the corresponding conclusion that there is no significant difference in those policies.

In Figure 9, it is not made clear that no data after the 20% level were used in making the predictions.

Response: Thanks for pointing this out. The prediction in Figure 9 is shown in dashed curves with the same colors as the reality for each variant. To make it clear, we add points (where the prediction starts) and a dash straight line at 10%, with corresponding illustration in the figure legend as "Figure 9. Estimated spread trend of Alpha, Delta, and Omicron in the UK using the Bayesian model, based on GISAID data during its early stage. The black dashed line suggests when each variant becomes dominating (proportion > 50%). The prediction is based on the data early stage of each VOC (proportion < 10%, below the orange dashed line), and we predict the trend of VOC after it reaches 10% (above the orange dashed line)."

Regarding “Conclusions” It is not clear that adding “communication among major countries and the pandemic policies in each country” added anything useful to the estimates. No analysis presented addressed what these data added.

Response: There is no doubt that the communications among major countries will influence the pandemic in each country, because passengers can obviously be the carrier of SARS-CoV-2 genomes by international travel. Correspondingly, forbidding on international travel and other policies, such as wearing mask and avoiding crowd gathering, will also help control the pandemic. Our model explicitly considers these factors and evaluates the effect of the 13 NPIs and vaccination. To eliminate the confusion that other readers may have as you, we add "The communication between countries is important to the pandemic in many aspects, for example, passengers can carry new variants to other countries by international travel. Therefore, the corresponding policies discussed in the next subsection should also be considered in the model" in Section 2.2.

Regarding “Discussions”, there should be better focused discussion of what could be causing the patterns presented in the results section. There are many unfocused and unsupported comments in this section that are not focused on the major methodologies and results of the paper. The reason why the Alpha vs. Delta comparisons in USA and UK are clearer than the comparisons with other VOC in Russia, India, and Indonesia may lie in the pre-Variants in those locations already had higher transmissibility or immunity escape capacities or because a greater fraction of the population had been affected by the pre-variants.

Response: Yes, you are correct that different pre-dominant variants will cause different environments for VOC to spread, and that is why most of our analysis is focusing on the comparison between VOC and the pre-dominant variant in each country. The results in Figure 3-5 are all the relative advantage of B.1.617.2, rather than the characteristic of VOC alone. Meanwhile, results of the UK and US are more reliable not because B.1.1.7 is the pre-dominant variant, but that they submit a large number of records of GISAID.

  • We add in the Discussion that "It is important to point out that this advantage is relative, rather than the characteristic of B.1.617.2 variant alone. This may explain the fact that Delta puts an end to the Alpha era and dominates the world in a few months".

  • Regarding to your comments in the Discussion section, we remove the following sentences that may cause readers' confusion:

    • "In Indonesia, the clock rate of B.1.617.2 is the highest, suggesting that its various mutations existed but all fail to have significant advantage over others, thus B.1.617.2 is still the dominating one. Ratio of reproductive numbers also indicates that the advantage in Indonesia is lowest (but still over 1)."

    • "Based on the results of clock rate, the advantage of B.1.617.2 to its pre-dominant variants is more obvious compared to that within Delta variants. The clock rate of the VOC is also found to be higher than the pre-dominant variant in Figure 4a and Figure 7a."

    • "When more data is available, the estimation of clock rate and reproductive number of Omicron will improve our understanding of this new VOC."

    • “We suggest government to develop new technology on testing and stricter rules on international travel to get the pandemic under better control. Policy on international travel policy will affect the relationship between countries as well, which should receive better effect on the VOC.”

    • “No significant difference in the policy effect is observed within Delta family, when comparing AY.4 and B.1.617.2.”

    • "Emergence of new variants and adjustment of anti-pandemic policies will affect the spread of VOC, contributing to the inconsistency in the late-stage data of each VOC."

    • "Accurate evaluation of policy effect on Omicron, whether it is too severe or too conservative, requires data for a longer period during which the policy changes and is conducted effectively."

The rationale for not choosing South Africa, where Omicron was first detected, is not sound. Where is the evidence that the local testing data is inaccurate? The data are probably comparable or better than Russia or Indonesia.

Response: According to another reviewer's suggestion, for the purpose of prediction of dominating date, we focus on the UK to conduct analysis on Alpha, Delta and Omicron, to eliminate the different situations between countries. The issue with South Africa is mainly the shortage of health system, as the president of South Africa, Cyril Ramaphosa, said, "the country faced a serious shortage of more than 12000 healthcare workers, including nurses, doctors and physiotherapists." Another paper from Nature points out that "Shortages of materials were initially the greatest limitation in fighting the pandemic on the continent... This is far below its needs, but many more than seemed possible when we launched PACT". If we compare the UK and South Africa, the UK has over 67 million population while South Africa's population is about 59 million. However, the UK conducts 1.02 million testings on 2021-12-01, while South Africa only conducts 35 thousand testing on the same day. This is during the expansion of Omicron which first started in South Africa, while its health system still fails to conduct enough number of testings. We revised the manuscript in Discussion as "Though it was first found in Africa, the statistics of local testing data is inaccurate due to lack of reagent test kit and compliance of masses. The population of the UK and South Africa in 2020 is roughly 67 and 59 million respectively, however, the daily testing number on 2021-12-01 is 1.8 million versus 35 thousand, suggesting that significant shortage of testing in South Africa".

The deletion in AY.4 may have helped focus sequencing on more AY.4 exposed than on B.1.617.2 exposed.

Response: Thank you for pointing this out. Firstly, according to the agreement of GISAID, "all users agree that no restrictions shall be attached to data submitted to GISAID, to promote collaboration among researchers on the basis of open sharing of data and respect for all rights and interests". Secondly, as both variants are from Delta family, before sequencing the researchers shall not know which variant the sample is. Therefore this should place little bias on our conclusion.

There is little base for the unhelpful comment: “We suggest government to develop new technology on testing and stricter rules on international travel to get the pandemic under better control. Policy on international travel policy will affect the relationship between countries as well, which should receive better effect on the VOC.” Since this is such a controversial issue, such comments should not be made without specific evidence. Policy in China on these issues should be different from that in other areas due to the low level of transmission that has been sustained in China.

Response: Thanks for your suggestion. We have deleted it in the revised version.
